# FlowTurbo: Towards Real-time Flow-Based Image Generation with Velocity Refiner

**Wenliang Zhao**[*]
Department of Automation
Tsinghua University
wenliangzhao.thu@gmail.com

**Minglei Shi**[*]
Department of Automation
Tsinghua University
stephenserrylei@gmail.com

**Xumin Yu**
Department of Automation
Tsinghua University
yuxumin98@gmail.com

**Jie Zhou**
Department of Automation
Tsinghua University
jzhou@tsinghua.edu.cn

**Jiwen Lu**
Department of Automation
Tsinghua University
lujiwen@tsinghua.edu.cn

## Abstract

Building on the success of diffusion models in visual generation, flow-based models reemerge as another prominent family of generative models that have achieved competitive or better performance in terms of both visual quality and inference speed. By learning the velocity field through flow-matching, flow-based models tend to produce a straighter sampling trajectory, which is advantageous during the sampling process. However, unlike diffusion models for which fast samplers are well-developed, efficient sampling of flow-based generative models has been rarely explored. In this paper, we propose a framework called FlowTurbo to accelerate the sampling of flow-based models while still enhancing the sampling quality. Our primary observation is that the velocity predictor's outputs in the flow-based models will become stable during the sampling, enabling the estimation of velocity via a lightweight velocity refiner. Additionally, we introduce several techniques including a pseudo corrector and sample-aware compilation to further reduce inference time. Since FlowTurbo does not change the multi-step sampling paradigm, it can be effectively applied for various tasks such as image editing, inpainting, *etc*. By integrating FlowTurbo into different flow-based models, we obtain an acceleration ratio of 53.1%∼58.3% on class-conditional generation and 29.8%∼38.5% on text-to-image generation. Notably, FlowTurbo reaches an FID of 2.12 on ImageNet with 100 (ms / img) and FID of 3.93 with 38 (ms / img), achieving the real-time image generation and establishing the new state-of-the-art. Code is available at https://github.com/shiml20/FlowTurbo.

## 1 Introduction

In recent years, diffusion models have emerged as powerful generative models, drawing considerable interest and demonstrating remarkable performance across various domains[10, 38, 30, 12]. Diffusion models utilize a denoising network, $\epsilon_\theta$, to learn the reverse of a diffusion process that gradually adds noise to transform the data distribution into a Gaussian distribution. While the formulation of diffusion models enables stable training and flexible condition injection[30], sampling from these models requires iterative denoising. This process necessitates multiple evaluations of the denoising network, thereby increasing computational costs. To address this, several techniques such as fast

---

[*]Equal contribution. [†]Corresponding author.

38th Conference on Neural Information Processing Systems (NeurIPS 2024).

diffusion samplers[22, 18, 42] and efficient distillation[31, 37] have been proposed to reduce the sampling steps of diffusion models.

Alongside the research on diffusion models, flow-based models[5, 19, 17] have garnered increasing attention due to their versatility in modeling data distributions. Flow is defined as a probability path that connects two distributions and can be efficiently modeled by learning a neural network to estimate the conditional velocity field through a neural network $\mathbf{v}_\theta$ via flow matching [17]. Encompassing the standard diffusion process as a special case, flow-based generative models support more flexible choices of probability paths. Recent work has favored a simple linear interpolant path [20, 24, 8], which corresponds to the optimal transport from the Gaussian distribution to the data distribution. This linear connection between data and noise results in a more efficient sampling process for flow-based models. However, unlike diffusion models, which benefit from numerous efficient sampling methods, current samplers for flow-based models primarily rely on traditional numerical methods such as Euler's method and Heun's method [24]. These traditional methods, while functional, fail to fully exploit the unique properties of flow-based generative models, thereby limiting the potential for faster and more efficient sampling.

In this paper, we propose FlowTurbo, a framework designed to accelerate the generation process of flow-based generative models. FlowTurbo is motivated by comparing the training objectives of diffusion and flow-based generative models, as well as analyzing how the prediction results $\epsilon_\theta$ and $\mathbf{v}_\theta$ vary over time. Our observation, illustrated in Figure 1, indicates that the velocity predictions of a flow-based model remain relatively stable during sampling, in contrast to the more variable predictions of $\epsilon_\theta$ in diffusion models. This stability allows us to regress the offset of the velocity at each sampling step using a lightweight velocity refiner, which contains only $5\%$ of the parameters of the original velocity prediction model. During the sampling process, we can replace the original velocity prediction model with our lightweight refiner at specific steps to reduce computational costs.

As a step towards real-time image generation, we propose two useful techniques called pseudo corrector and sample-aware compilation to further improve the sampling speed. Specifically, the pseudo corrector method modifies the updating rule in Heun's method by reusing the velocity prediction of the previous sampling step, which will reduce the number of model evaluations at each step by half while keeping the original convergence order. The sample-aware compilation integrates the model evaluations, the sampling steps as well as the classifier-free guidance [11] together and compile them into a static graph, which can bring extra speedup compared with standard model-level compilation. Since each sample block is independent, we can still adjust the number of inference steps and sampling configurations flexibly.

Our FlowTurbo framework is fundamentally different from previous one-step distillation methods for diffusion models [20, 40, 32], which require generating millions of noise-image pairs offline and conducting distillation over hundreds of GPU days. In contrast, FlowTurbo's velocity refiner can be efficiently trained on pure images in less than 6 hours. Moreover, one-step distillation-based methods are limited to image generation and disable most of the functionalities of the original base model. Conversely, FlowTurbo preserves the multi-step sampling paradigm, allowing it to be effectively applied to various tasks such as image editing, inpainting, and more.

We perform extensive experiments to evaluate our method. By applying FlowTurbo to different flow-based models, we obtain an acceleration ratio of 53.1%∼58.3% on class-conditional generation and 29.8%∼38.5% on text-to-image generation. Notably, FlowTurbo attains an FID score of 2.12 on ImageNet with 100 (ms / img) and FID of score 3.93 with 38 (ms / img), thereby enabling real-time image generation and establishes the new state-of-the-art. Additionally, we present qualitative comparisons demonstrating how FlowTurbo generates superior images with higher throughput and how it can be seamlessly integrated into various applications such as image editing, inpainting, *etc*. We believe our FlowTurbo can serve as a general framework to accelerate flow-based generative models and will see wider use as these models continue to grow [24, 20, 8, 9].

## 2 Related Work

**Diffusion and flow-based models.** Diffusion models [10, 38] are a family of generative models that have become the de-facto method for high-quality generation. The diffusion process gradually adds noise to transform the data distribution to a normal distribution, and the goal of diffusion models is to use a network $\epsilon_\theta$ to learn the reverse of the diffusion process via score-matching [10, 38]. Rombach *et*

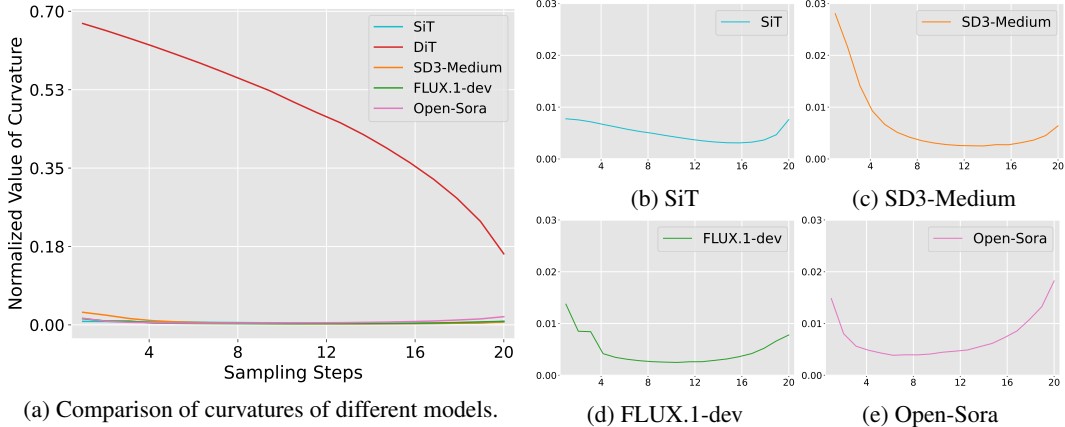

(a) Comparison of curvatures of different models.

(b) SiT

(c) SD3-Medium

(d) FLUX.1-dev

(e) Open-Sora

Figure 1: **Visualization of the curvatures of the sampling trajectories of different models.** We compare the curvatures of the model predictions of a standard diffusion model (DiT [28]) and several flow-based models (SiT [24], SD3-Medium [8], FLUX.1-dev [14], and Open-Sora [43]) during the sampling. We observe that the $\mathbf{v}_\theta$ in flow-based models is much more stable than $\epsilon$ of diffusion models during the sampling, which motivates us to seek a more lightweight estimation model to reduce the sampling costs of flow-based generative models.

*al.* [30] first scales up diffusion models to large-scale text-to-image generation by performing the diffusion on latent space and adopting cross-attention to inject conditions. The pre-trained diffusion models can also be easily fine-tuned to achieve generation with more diverse conditions [41, 27] and have attracted increasing attention in the community. Flow-based generative models are different from diffusion models in both data modeling and training objectives. Flow-based models [20, 17, 8, 24] consider the probability path from one distribution to another, and learn the velocity field via flow matching [17]. By choosing the linear interpolant as the probability path which corresponds to the optimal transport from the normal distribution to the data distribution, the trajectory from noise to data becomes more straighter which is beneficial to the sampling. Recent work [24, 8] have demonstrates the effectiveness and scalability of flow-based generation models. However, both diffusion and flow-based models requires multiple evaluations of the prediction model, leading to lower inference speed than traditional architectures like GAN. In this work, we focus on this issue and aim to accelerate flow-based generative models.

**Efficient visual generation.** Accelerating the generation of diffusion models has become an increasingly important topic. Existing methods can be roughly categorized as training-free and training-based methods. Training-free methods aim to design faster samplers that can reduce the approximation error when sampling from the diffusion SDE or ODE [36, 22, 18, 42], while keeping the weights of diffusion models unchanged. Training-based methods often aim to reshape the sampling trajectory by distillation from the diffusion model [31, 40] to achieve the few-step or even one-step generation. These training-based methods usually requires multiple-round of distillation [31, 20] and expensive training resources (*e.g.*, >100 GPU days in [20]). Besides, the distilled one-step model no longer supports image editing due to the lack of multi-step sampling. Although there are a variety of methods for accelerating diffusion models, there are few fast sampling methods designed for flow-based generative models. Existing flow-based models adopt traditional numerical methods like Euler's method or Heun's method during the inference [24]. In this work, we provide a framework called FlowTurbo to accelerate the generation of flow-based models by learning a lightweight velocity refiner (which only requires <6 GPU hours) to regress the offset of the velocity. Together with other proposed techniques, FlowTurbo addresses the previously unmet need for an efficient flow-based generation framework, paving the way for real-time generative applications.

## 3 Method

### 3.1 Preliminaries: Diffusion and Flow-based Models

**Diffusion models.** Recently, diffusion models [10, 38, 35, 30] have emerged as a powerful family of generative models. The diffusion models are trained to learn the inverse of a diffusion process such

that it can recover the data distribution $p_0(\mathbf{x}_0)$ from the Gaussian noise. The diffusion process can be represented as:

$$\mathbf{x}_t = \alpha_t \mathbf{x}_0 + \sigma_t \boldsymbol{\epsilon}, \quad t \in [0,1], \quad \boldsymbol{\epsilon} \sim \mathcal{N}(0, \mathbf{I}), \tag{1}$$

where $\alpha_t, \sigma_t$ are the chosen noise schedule such that the marginal distribution $p_1(\mathbf{x}_1) \sim \mathcal{N}(0, \mathbf{I})$. The optimization of diffusion models can be derived by either minimizing the ELBO of the reverse process [10] or solving the reverse diffusion SDE [38], which would both lead to the same training objective of score-matching, *i.e.*, to learn a noise prediction model $\boldsymbol{\epsilon}_\theta(\mathbf{x}_t, t)$ to estimate the scaled score function $-\sigma_t \nabla_\mathbf{x} \log p_t(\mathbf{x}_t)$:

$$\mathcal{L}_{\mathrm{DM}}(\theta) = \mathbb{E}_{t, p_0(\mathbf{x}_0), p(\mathbf{x}_t | \mathbf{x}_0)} \left[ \lambda(t) \left\| \boldsymbol{\epsilon}_\theta(\mathbf{x}_t, t) + \sigma_t \nabla_\mathbf{x} \log p_t(\mathbf{x}_t) \right\|_2^2 \right], \tag{2}$$

where $\lambda(t)$ is a time-dependent coefficient. Sampling from a diffusion model can be achieved by solving the reverse-time SDE or the corresponding diffusion ODEs [38], which can be efficiently achieved by modern fast diffusion samplers [36, 22, 42].

**Flow-based models.** Flow-based models can be traced back to Continuous Normalizing Flows [5] (CNF), which is a more generic modeling technique and can capture the probability paths of the diffusion process as well [17]. Training a CNF becomes more practical since the purpose of the flow matching technique [17], which learns the conditional velocity field of the flow. Similar to (1), we can add some constraints to the noise schedule such that $\alpha_0 = 1, \sigma_0 = 0$ and $\alpha_1 = 0, \sigma_1 = 1$, and then define the flow as:

$$\psi_t(\cdot | \boldsymbol{\epsilon}) : \mathbf{x}_0 \mapsto \alpha_t \mathbf{x}_0 + \sigma_t \boldsymbol{\epsilon}, \tag{3}$$

In this case, the velocity field that generates the flow $\psi_t$ can be represented as:

$$u_t(\psi_t(\mathbf{x}_0 | \boldsymbol{\epsilon}) | \boldsymbol{\epsilon}) = \frac{\mathrm{d}}{\mathrm{d}t} \psi_t(\mathbf{x}_0 | \boldsymbol{\epsilon}) = \dot{\alpha}_t \mathbf{x}_0 + \dot{\sigma}_t \boldsymbol{\epsilon}. \tag{4}$$

The training objective of conditional flow matching is to train a velocity prediction model $\mathbf{v}_\theta$ to estimate the conditional velocity field:

$$\mathcal{L}_{\mathrm{FM}}(\theta) = \mathbb{E}_{t, p_1(\boldsymbol{\epsilon}), p_0(\mathbf{x}_0)} \left\| \mathbf{v}_\theta(\psi_t(\mathbf{x}_0 | \boldsymbol{\epsilon}), t) - \frac{\mathrm{d}}{\mathrm{d}t} \psi_t(\mathbf{x}_0 | \boldsymbol{\epsilon}) \right\|_2^2 \tag{5}$$

The sampling of a flow-based model can be achieved by solving the probability flow ODE with the learned velocity

$$\frac{\mathrm{d}\mathbf{x}_t}{\mathrm{d}t} = \mathbf{v}_\theta(\mathbf{x}_t, t), \quad \mathbf{x}_1 \sim p_1(\mathbf{x}_1). \tag{6}$$

Since the formulation of the flow $\psi_t$ can be viewed as the interpolation between $\mathbf{x}_0$ and $\mathbf{v}$, it is also referred to as interpolant in some literature [1, 24]. Among various types of interpolants, a very simple choice is linear interpolant [24, 8], where $\alpha_t = (1 - t)$ and $\sigma_t = t$. In this case, the velocity field becomes a straight line connecting the initial noise and the data point, which also corresponds to the optimal transport between the two distributions [19, 17]. The effectiveness and scalability of the linear interpolant have also been proven in recent work [17, 24, 8].

### 3.2 Efficient Estimation of Velocity

We consider the velocity estimation in flow-based generative models with the linear interpolant [24, 8, 20]. As shown in (5), the training target of the velocity prediction model $\mathbf{v}_\theta$ is exactly $\boldsymbol{\epsilon} - \mathbf{x}_0$, a constant value independent of $t$. Our main motivation is to efficiently estimate the velocity during the sampling, instead of evaluating the whole velocity prediction model $\mathbf{v}_\theta$ every time.

**Analyzing the stability of velocity.** We start by analyzing the stability of the output value of $\mathbf{v}_\theta$ along the sampling trajectory. By comparing the training objectives of diffusion and flow-based models (2)(5), we know that the target of $\mathbf{v}_\theta$ is independent of $t$. A more in-depth discussion is provided in Appendix A.3, where we show the two training objectives have different time-dependent weight functions. To verify whether there are similar patterns during the sampling, we compare how the prediction results change across the sampling steps in Figure 1. Specifically, we compare the curvatures of $\boldsymbol{\epsilon}_\theta$ of a diffusion model (DiT [28]) and the $\mathbf{v}_\theta$ of flow-based models (SiT [8], SD3 [8], etc) during the sampling steps. For each model, we sample from 8 random noises and set the total sampling steps as 20. It can be clearly observed that $\mathbf{v}_\theta$ of a flow-based model is much more stable

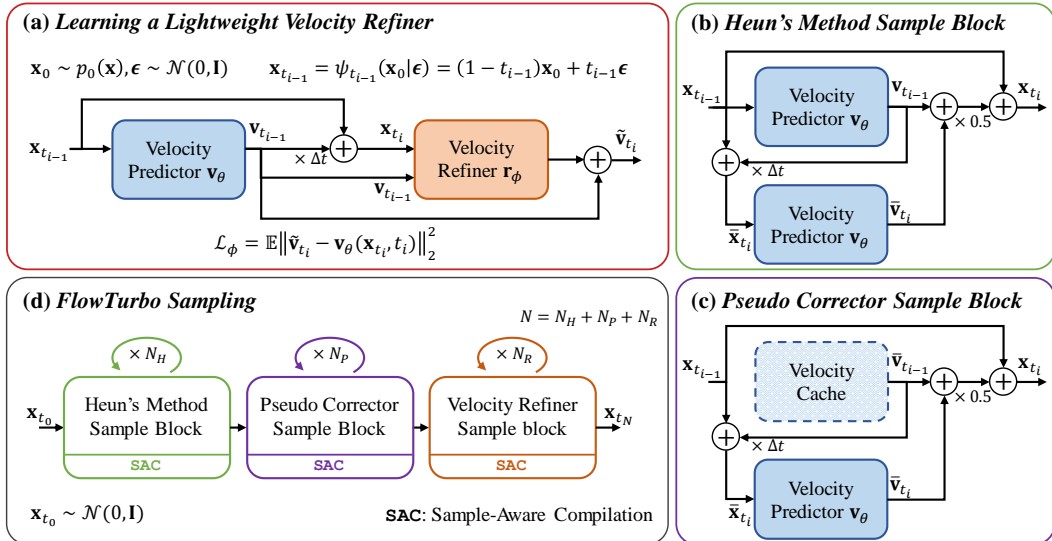

Figure 2: **Overview of FlowTurbo.** **(a)** Motivated by the stability of the velocity predictor's outputs during the sampling, we propose to learn a lightweight velocity refiner to regress the offset of the velocity field. **(b)(c)** We propose the pseudo corrector which leverages a velocity cache to reduce the number of model evaluations while maintaining the same convergence order as Heun's method. **(d)** During sampling, we employ a combination of Heun's method, the pseudo corrector, and the velocity refiner, where each sample block is processed with the proposed sample-aware compilation.

than the $\epsilon_\theta$ of a diffusion model. Therefore, We define the $\mathbf{v}_\theta$ as a *"stable value"*. The stability of $\mathbf{v}_\theta$ makes it possible to obtain the velocity more efficiently rather than performing the forward pass of the whole velocity prediction network $\mathbf{v}_\theta$ at every sampling step.

**Learning a lightweight velocity refiner.** Since the velocity in a flow-based model is a "stable value", we propose to learn a lightweight refiner that can adjust the velocity with minimal computational costs. The velocity refiner takes as inputs both the current intermediate result and the velocity of the previous step, and returns the offset of velocity:

$$\mathbf{v}_{t_i} = \mathbf{r}_\phi(\mathbf{x}_{t_i}, \mathbf{v}_{t_{i-1}}, t_i) + \mathbf{v}_{t_{i-1}}. \tag{7}$$

The velocity refiner $\mathbf{r}_\phi$ can be designed to be very lightweight ($< 5\%$ parameters of $\mathbf{v}_\theta$). The detailed architecture can be found in Appendix C.

To learn the velocity refiner, we need to minimize the difference between the output of $\mathbf{r}_\phi$ and the actual offset $\mathbf{v}_{t_i} - \mathbf{v}_{t_{i-1}}$. However, it requires multiple-step sampling to obtain an intermediate result $\mathbf{x}_{t_i}$ to make the training objective perfectly align with our target. To reduce the training cost, we simulate the $\mathbf{x}_t$ with one-step sampling starting from $\mathbf{x}_{t_{i-1}}$, which is directly obtained by the flow $\psi_{t_{i-1}}$. The detailed procedure to compute the loss is listed as follows:

$$\mathbf{x}_{t_{i-1}} \leftarrow \psi_{t_{i-1}}(\mathbf{x}_0|\boldsymbol{\epsilon}), \quad \mathbf{x}_0 \sim p_0(\mathbf{x}), \boldsymbol{\epsilon} \sim p_1(\mathbf{x}) \tag{8}$$

$$\mathbf{v}_{t_{i-1}} \leftarrow \mathbf{v}_\theta(\mathbf{x}_{t_{i-1}}, t_{i-1}), \quad \mathbf{x}_{t_i} \leftarrow \text{Solver}(\mathbf{x}_{t_{i-1}}, \mathbf{v}_{t_{i-1}}, \Delta t) \tag{9}$$

$$\mathcal{L}_\phi \leftarrow \mathbb{E}\|\mathbf{v}_\theta(\mathbf{x}_{t_i}, t_i) - (\mathbf{r}_\phi(\mathbf{x}_{t_i}, \mathbf{v}_{t_{i-1}}, t_i) + \mathbf{v}_{t_{i-1}})\|_2^2 \tag{10}$$

Where $\Delta t = t_i - t_{i-1}$ and we use a simple Euler step for the Solver to obtain $\mathbf{x}_{t_i}$. Once the velocity refiner is learned, we can use it to replace the original $\mathbf{v}_\theta$ at some specific sampling steps. We will demonstrate through experiments that adding the velocity refiner can improve the sampling quality without introducing noticeable computational overhead.

**Compatibility with classifier-free guidance.** Classifier-free guidance [11] is a useful technique to improve the sampling quality in conditional sampling. Let $\mathbf{y}$ be the condition, the classifier-free guidance for a velocity prediction model [8] can be defined as:

$$\mathbf{v}^\zeta(\mathbf{x}, t|\mathbf{y}) = (1 - \zeta)\mathbf{v}_\theta(\mathbf{x}, t|\varnothing) + \zeta\mathbf{v}_\theta(\mathbf{x}, t|\mathbf{y}), \tag{11}$$

where $\zeta$ is the guidance scale and $\varnothing$ denotes the null condition. To make our velocity refiner support classifier-free guidance, we only need to make sure both the conditional prediction $\mathbf{v}_\theta(\mathbf{x}, t|\mathbf{y})$ and the

unconditional prediction $\mathbf{v}_\theta(\mathbf{x}, t|\varnothing)$ appear during the training. Note that we always feed the velocity prediction model $\mathbf{v}_\theta$ and the velocity refiner $\mathbf{r}_\phi$ with the same condition.

$$\mathbf{y}_\gamma = \mathbb{I}_{\gamma \le \gamma_1} \cdot \varnothing + \mathbb{I}_{\gamma > \gamma_1} \cdot \mathbf{y}, \quad \gamma \in \mathcal{U}[0, 1], \tag{12}$$

$$\mathcal{L}_\phi^{\mathrm{CFG}} \leftarrow \mathbb{E}\|\mathbf{v}_\theta(\mathbf{x}_{t_i}, t_i|\mathbf{y}_\gamma) - (\mathbf{r}_\phi(\mathbf{x}_{t_i}, \mathbf{v}_{t_{i-1}}, t_i|\mathbf{y}_\gamma) + \mathbf{v}_{t_{i-1}})\|_2^2, \tag{13}$$

where we set the $\gamma_1 = 0.1$ as the probability of using an unconditional velocity.

### 3.3 Towards Real-Time Image Generation

The sampling costs of a flow-based model can be significantly minimized by integrating our lightweight velocity refiner $\mathbf{r}_\phi$ in place of the velocity prediction network $\mathbf{v}_\theta$ at selected sampling steps. In this section, we propose two techniques to further improve the sampling speed towards real-time image generation.

**Pseudo corrector.** Traditional numerical ODE solvers are usually used to sample from a probability flow ODE. For example, SiT [8] adopt a Heun method (or improved Euler's method) [15] as the ODE solver. The update rule from $t_{i-1}$ to $t_i$ can be written as:

$$\mathbf{d}_{i-1} \leftarrow \mathbf{v}_\theta(\mathbf{x}_{t_{i-1}}, t_{i-1}|\mathbf{y}), \qquad \tilde{\mathbf{x}}_{t_i} \leftarrow \mathbf{x}_{t_{i-1}} + \Delta t \mathbf{d}_{i-1} \tag{14}$$

$$\mathbf{d}_i \leftarrow \mathbf{v}_\theta(\tilde{\mathbf{x}}_{t_i}, t_i|\mathbf{y}), \qquad \mathbf{x}_i \leftarrow \mathbf{x}_{i-1} + \frac{\Delta t}{2}[\mathbf{d}_{i-1} + \mathbf{d}_i] \tag{15}$$

Each Heun step contains a predictor step (14) and a corrector step (15), thus includes two evaluations of the velocity predictor $\mathbf{v}_\theta$, bringing extra inference costs. Motivated by [42], we propose to reuse the $\mathbf{d}_i$ in the next sampling step, instead of re-computing it via $\mathbf{d}_i \leftarrow \mathbf{v}_\theta(\mathbf{x}_{t_i}, t_i|\mathbf{y})$ (see Figure 2 (b)(c) for illustration). We call this a pseudo corrector since it is different from the predictor-corrector solvers in numerical analysis. It can be proved (see Appendix B) that the pseudo corrector also enjoys 2-order convergence while only having one model evaluation at each step.

**Sample-aware compilation.** Compiling the network into a static graph is a widely used technique for model acceleration. However, all the previous work only considers network-level compilation, *i.e.*, only compiling the $\epsilon_\theta$ or $\mathbf{v}_\theta$. We propose the sample-aware compilation which wraps both the forward pass of $\mathbf{v}_\theta$ or $\mathbf{r}_\phi$ and the sampling operation together (including the classifier-free guidance) and performs the compilation. For example, the sample blocks illustrated in Figure 2 (b, c) are compiled into static graphs. Since each sample block is independent, we can still adjust the number of inference steps and sampling configurations flexibly.

### 3.4 Discussion

Recently, there have been more and more training-based methods [20, 40, 32] aiming to accelerate diffusion models or flow-based models through one-step distillation. Although these methods can achieve faster inference, they usually require generating paired data using the pre-trained model and suffer from large training costs (*e.g.*, >100 GPU days in [40, 20]). Besides, one-step methods only keep the generation ability of the original model while disabling more diverse applications such as image inpainting and image editing. In contrast, our FlowTurbo aims to accelerate flow-based models through velocity refinement, which still works in a multi-step manner and performs sampling on the original trajectory. For example, FlowTurbo can be easily combined with existing diffusion-based image editing methods like SDEdit [25] (see Section 4.4).

## 4 Experiments

We conduct extensive experiments to verify the effectiveness of FlowTurbo. Specifically, we apply FlowTurbo to both class-conditional image generation and text-to-image generation tasks and demonstrate that FlowTurbo can significantly reduce the sampling costs of the flow-based generative models. We also provide a detailed analysis of each component of FlowTurbo, as well as qualitative comparisons of different tasks.

### 4.1 Setups

In our experiments, we consider two widely used benchmarks including class-conditional image generation and text-to-image generation. For class-conditional image generation, we adopt a transformer-

Table 1: **Main results.** We apply our FlowTurbo on SiT-XL [24] and the 2-RF of InstaFlow [20] to perform class-conditional image generation and text-to-image generation, respectively. The image quality is measured by the FID 50K↓ on ImageNet (256×256) and the FID 5K↓ on MS COCO 2017 (512×512). We use the suffix to represent the number of Heun's method block ($H$), pseudo corrector block ($P$), and the velocity refiner block ($R$). Our results demonstrate that FlowTurbo can significantly accelerate the inference of flow-based models while achieving better sampling quality.

(a) Class-conditional Image Generation

| Method | Sample Config | FLOPs (G) | FID↓ | Latency (ms / img) |
|---|---|---|---|---|
| *SiT-XL [8], ImageNet* (256 × 256) | | | | |
| Heun's | $H_8$ | 1898 | 3.68 | 89.4 |
| FlowTurbo | $H_2P_4R_2$ | 957 | 3.63 | 41.6 (-53.4%) |
| Heun's | $H_{11}$ | 2610 | 2.79 | 117.8 |
| FlowTurbo | $H_2P_8R_2$ | 1431 | 2.69 | 55.2 (-53.1%) |
| Heun's | $H_{15}$ | 3559 | 2.42 | 154.8 |
| FlowTurbo | $H_5P_7R_3$ | 2274 | 2.22 | 72.5 (-53.2%) |
| Heun's | $H_{24}$ | 5694 | 2.20 | 240.6 |
| FlowTurbo | $H_8P_9R_5$ | 3457 | 2.12 | 100.3 (-58.3%) |

(b) Text-to-image Generation

| Method | Sample Config | FLOPs (G) | FID↓ | Latency (ms / img) |
|---|---|---|---|---|
| *InstaFlow [20], MS COCO 2017* (512 × 512) | | | | |
| Heun's | $H_4$ | 3955 | 32.77 | 104.5 |
| FlowTurbo | $H_1P_2R_2$ | 2649 | 32.48 | 68.4 (-34.5%) |
| Heun's | $H_5$ | 4633 | 30.73 | 120.3 |
| FlowTurbo | $H_1P_4R_2$ | 3327 | 30.19 | 84.5 (-29.8%) |
| Heun's | $H_8$ | 6667 | 28.61 | 170.5 |
| FlowTurbo | $H_1P_6R_3$ | 4030 | 28.60 | 104.8 (-38.5%) |
| Heun's | $H_{10}$ | 8023 | 28.06 | 203.7 |
| FlowTurbo | $H_3P_6R_3$ | 5386 | 27.60 | 137.0 (-32.7%) |

Table 2: **Comparisons with the state-of-the-arts**. We compare the sampling quality and speed of different methods on ImageNet 256 × 256 class-conditional sampling. We demonstrate that FlowTurbo can significantly improve over the baseline SiT-XL [24] and achieves the fastest sampling (38 ms / img) and the best quality (2.12 FID) with different configurations.

| Model | Sample Config | Params | Latency (ms / img) | FID↓ | IS↑ | Precision↑ | Recall↑ |
|---|---|---|---|---|---|---|---|
| StyleGAN-XL [33] | - | 166M | 190 | 2.30 | 265.1 | 0.78 | 0.53 |
| Mask-GIT [2] | 8 steps | 227M | 120 | 6.18 | 182.1 | 0.80 | 0.51 |
| ADM [7] | 250 steps DDIM [36] | 554M | 2553 | 10.94 | 101.0 | 0.69 | **0.63** |
| ADM-G [7] | 250 steps DDIM [36] | 608M | 4764 | 4.59 | 186.7 | 0.83 | 0.53 |
| LDM-4-G [30] | 250 steps DDIM [36] | 400M | 448 | 3.60 | 247.7 | **0.87** | 0.48 |
| DiT-XL [28] | 250 steps DDPM [10] | 675M | 6914 | 2.27 | **278.2** | 0.83 | 0.57 |
| SiT-XL [24] | 25 steps dopri5 [15] | 675M | 3225 | 2.15 | 258.1 | 0.81 | 0.60 |
| SiT-XL [24] | 25 steps Heun's [15] | 675M | 250 | 2.20 | 254.9 | 0.81 | 0.60 |
| FlowTurbo (**ours**) | $H_1P_5R_3$ | 704M | **38** | 3.93 | 223.6 | 0.79 | 0.56 |
| FlowTurbo (**ours**) | $H_5P_7R_3$ | 704M | 73 | 2.22 | 248.0 | 0.81 | 0.60 |
| FlowTurbo (**ours**) | $H_8P_9R_5$ | 704M | 100 | **2.12** | 255.6 | 0.81 | 0.60 |

style flow-based model SiT-XL [24] pre-trained on ImageNet 256×256. For text-to-image generation, we utilize InstaFlow [20] as the flow-based model, whose backbone is a U-Net similar to Stable-Diffusion [30]. Note that we use the 2-RF model from [19] instead of the distilled version since our FlowTurbo is designed to achieve acceleration within the multi-step sampling framework. The velocity refiner only contains 4.3% and 5% parameters of the corresponding predictor, and the detailed architecture can be found in Appendix C. During training, we randomly sample $\Delta t \in (0, 0.12]$ and compute the training objectives in (13). In both tasks, we use a single NVIDIA A800 GPU to train the velocity refiner and find it converges within 6 hours. We use a batch size of 8 on a single A800 GPU to measure the latency of each method. Please refer to Appendix C for more details.

## 4.2 Main Results

**Class-conditional image generation.** We adopt the SiT-XL [24] trained on ImageNet [6] of resolution of 256 × 256. Following common practice [24, 30], we adopt a classifier-free guidance scale (CFG) of 1.5. According to [24], a widely used sampling method of the flow-based model is Heun's method [15]. In Table 1a, we demonstrate how our FlowTurbo can achieve faster inference than Heun's method in various computational budgets. Specifically, we conduct experiments with different sampling configurations (the second column of Table 1a), where we use the suffix to represent the number of Heun's method block ($H$), pseudo corrector block ($P$), and the velocity refiner block ($R$). Note that each Heun's block contains two evaluations of the velocity predictor while each pseudo corrector block only contains one. We also provide the total FLOPs during the sampling and the

inference speed of each sample configuration. In each group of comparison, we choose the sampling strategy of FlowTurbo to make the sampling quality (measured by the FID 50K↓) similar to the baseline. Our results demonstrate that FlowTurbo can accelerate the inference by $37.2\% \sim 43.1\%$, while still achieving better sampling quality. Notably, FlowTurbo obtains 3.63 FID with a sampling speed of 41.6 ms/img, achieving real-time image generation.

**Text-to-image generation.** We adopt the 2-RF model in [20] as our base model for text-to-image generation. Note that we do not adopt the distilled version in [20] since we focus on accelerating flow-based models within the multi-step sampling paradigm. Following [20, 26], we compute the FID 5K↓ between the generated $512 \times 512$ samples and the images on MS COCO 2017 [16] validation set. The results are summarized in Table 1b, where we compare the sampling speed/quality with the baseline Heun's method. Note that the notation of the sampling configuration is the same as Table 1a. The results clearly demonstrate that Our FlowTurbo can also achieve significant acceleration (29.8%~38.5%) on text-to-image generation.

## 4.3 Comparisons to State-of-the-Arts

In Table 2, we compare our FlowTurbo with state-of-the-art methods on ImageNet $256 \times 256$ class-conditional generation. We use SiT-XL [24] as our base model and apply FlowTurbo with different sampling configurations on it. We show that FlowTurbo with $H_1 P_5 R_3$ achieves the sampling speed of 38 (ms / img) with 3.93 FID (still better than most methods like Mask-GIT [2], ADM [7]). On the other hand, FlowTurbo with $H_8 P_9 R_5$ archives the lowest FID 2.12, outperforming all the other methods. Besides, we also provide a comparison of the sampling speed/quality trade-offs of SiT (by changing the number of sampling steps of Heun's method) and FlowTurbo (by changing the sampling configurations) in 3, where the results of some other state-of-the-arts methods are also included. The comparison shows our FlowTurbo exhibits favorable sampling quality/speed trade-offs.

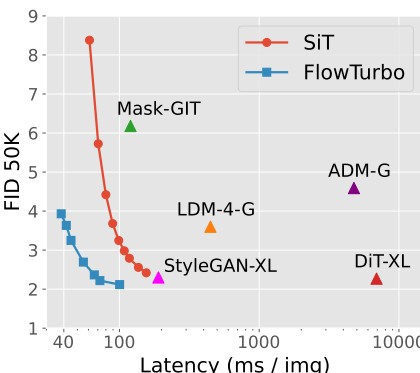

Figure 3: **FlowTurbo exhibits favorable trade-offs compared with SOTA methods.**

## 4.4 Analysis

**Ablation of components of FlowTurbo.** We evaluate the effectiveness of each component of FlowTurbo in Table 3a. Specifically, we start from the baseline, a 7-step Heun's method and gradually add components of FlowTurbo. In the sample config A, we show that adding a velocity refiner can significantly improve the FID↓ ($4.42 \rightarrow 2.80$), while introducing minimal computational costs (only $+0.7\%$ in the latency). From B to E, we adjust the ratios of Heun's method block, the pseudo corrector block, and the velocity refiner block to achieve different trade-offs between sampling speed and quality. In the last two rows, we show that our sample-aware compilation is better than standard model-level compilation, further increasing the sampling speed.

**Choice of $\Delta t$.** We find the choice of $\Delta t$ during training is crucial and affects the sampling results a lot in our experiments, as shown in Table 3b. We find $\Delta t \in (0.0, 0.1]$ works well for more sampling steps like $H_{12} R_5$, while $\Delta t \in [0.06, 0.12]$ is better for fewer sampling steps like $H_6 R_2$ and $H_9 R_3$. Besides, we find $\Delta t \in (0.0, 0.12]$ yields relatively good results in all the situations.

**Effects of velocity refiner.** We evaluate the effects of the different number of velocity refiners in Table 3c, and find that appropriately increasing the number of velocity refiners can improve the trade-off between sampling quality and speed. Specifically, we find $H_6 R_2$ can achieve better image quality and generation speed than the baseline $H_8$.

**Effects of pseudo corrector.** In Table 3d, we fix the total number of both Heun's sample block and pseudo corrector block and adjust the ratio of the pseudo corrector. Our results demonstrate that increasing the number of pseudo corrector blocks can significantly improve the sampling speed while introducing neglectable performance drop (*e.g.*, FlowTurbo with $H_1 P_6 R_2$ performs better than $H_8$).

Table 3: **Ablation studies.** We evaluate the effectiveness of each component in FlowTurbo as well as the selection of some hyper-parameters. **(a)** We gradually add the components of FlowTurbo to the baseline and show that FlowTurbo can achieve over $50\%$ acceleration with better sampling quality. **(b)** we experiment with different ranges of $\Delta t$ and find $\Delta t \in (0.0, 0.12]$ yields relatively good results in all the situations. **(c)(d)** we show how the sampling quality/speed changes with the number of velocity refiner and pseudo corrector blocks.

(a) Ablation of components of FlowTurbo.

| Sample Config | $N_H$ | $N_P$ | $H_R$ | FID↓ | Latency (ms / img) |
|---|---|---|---|---|---|
| baselines | 7 | 0 | 0 | 4.42 | 80.0 |
| | 8 | 0 | 0 | 3.68 | 89.4 (+11.8%) |
| A | 7 | 0 | 1 | 2.80 | 80.5 (+0.7%) |
| B | 2 | 8 | 2 | 2.69 | 71.6 (-10.4%) |
| C | 3 | 3 | 2 | 3.25 | 57.4 (-28.2%) |
| D | 2 | 4 | 2 | 3.63 | 52.7 (-34.1%) |
| E | 1 | 5 | 3 | 3.93 | 48.0 (-40.0%) |
| E + Model-Level Comp. | | | | 3.93 | 41.0 (-48.7%) |
| E + Sample-Aware Comp. | | | | 3.93 | 38.3 (-52.2%) |

(c) Effects of the velocity refiner.

| Sample Config | FID↓ | Latency (ms / img) |
|---|---|---|
| $H_8$ | 3.68 | 68.0 |
| $H_7R_1$ | 2.80 | 61.6 (-9.4%) |
| $H_6R_2$ | 3.55 | 55.2 (-18.8%) |
| $H_5R_3$ | 7.62 | 49.9 (-26.5%) |

(d) Effects of the pseudo corrector.

| Sample Config | FID↓ | Latency (ms / img) |
|---|---|---|
| $H_8$ | 3.68 | 68.0 |
| $H_7R_2$ | 2.93 | 62.0 (-8.8%) |
| $H_6P_1R_2$ | 2.60 | 58.6 (-13.8%) |
| $H_5P_2R_2$ | 2.66 | 55.2 (-18.8%) |
| $H_4P_3R_2$ | 2.78 | 51.8 (-23.8%) |
| $H_3P_4R_2$ | 2.96 | 48.4 (-28.8%) |
| $H_2P_5R_2$ | 3.21 | 45.0 (-33.7%) |
| $H_1P_6R_2$ | 3.59 | 41.6 (-38.7%) |

(b) Ablation of $\Delta t$.

| Sample Config | $\Delta t$ \ FID↓ | | | |
|---|---|---|---|---|
| | (0.0, 0.1] | (0.0, 0.12] | (0.0, 0.2] | [0.06, 0.12] |
| $H_6R_2$ | 3.58 | 3.55 | 4.48 | **3.36** |
| $H_9R_3$ | 2.73 | 2.65 | 2.93 | **2.62** |
| $H_{12}R_5$ | **2.53** | 2.54 | 2.64 | 2.89 |

Table 4: **Comparisons of different orders of the blocks.** We compare the results of changing the orders of Heun's block ($H$), pseudo corrector block ($P$), and the velocity refiner block ($R$). Our results show applying the blocks in $H_{N_H} P_{N_P} R_{N_R}$ order yields the best trade-off between generation quality and speed.

(a) Altering the order of $H/P/R$ blocks.

| Sample Config | FID ↓ | Latency (ms / img) |
|---|---|---|
| $H_5P_7R_3$ | **2.24** | 72.4 |
| $P_5H_7R_3$ | 2.38 | 79.2 |
| $P_5R_7H_3$ | 13.66 | 53.7 |
| $H_5R_7P_3$ | 30.94 | 60.4 |

(b) Repeating multiple $H/P/R$ blocks

| Sample Config | FID ↓ | Latency (ms / img) |
|---|---|---|
| $[H_1P_1R_1]_{\times 2}$ | 8.15 | 34.8 |
| $H_2P_2R_2$ | 6.39 | 34.8 |
| $[H_1P_1R_1]_{\times 3}$ | 4.34 | 45.4 |
| $H_3P_3R_3$ | 3.34 | 45.4 |

**Ablation of refiner architectures.** When designing the architecture for the velocity refiner, we followed a simple rule to make the refiner have a similar architecture as the base velocity predictor but with much fewer parameters ( 5% of the base model). The detailed architecture is described in Section 4.1 and Appendix C. For example, since SiT consists of multiple transformer blocks, we simply use a single block as the refiner. For text-to-image generation, we reduce the number of layers and channels of the UNet. In our early experiments, we have tried another architecture for class-conditional image generation, where a SiT-S (a smaller version of the base velocity predictor SiT-XL) is adopted as the refiner (as shown in Table 5). We find that using a block of SiT-XL as the refiner is slightly better than the SiT-S. These results demonstrate that our framework is robust to the choice of model architectures for the velocity refiner.

Table 5: **Ablation of refiner architectures.**

| Sample Config | Architecture | Params | FID ↓ |
|---|---|---|---|
| $H_5P_7R_3$ | SiT-S | 33M | 2.53 |
| $H_5P_7R_3$ | a block of SiT-XL | 29M | 2.22 |
| $H_8P_9R_5$ | SiT-S | 33M | 2.24 |
| $H_8P_9R_5$ | a block of SiT-XL | 29M | 2.12 |

**Comparisons of different order of the blocks.** According to the observation in Figure 1, the velocity during the sampling would become stable at the final few steps, where we adopted a lightweight refiner to regress the velocity offset. Besides, our pseudo corrector is designed to efficiently achieve

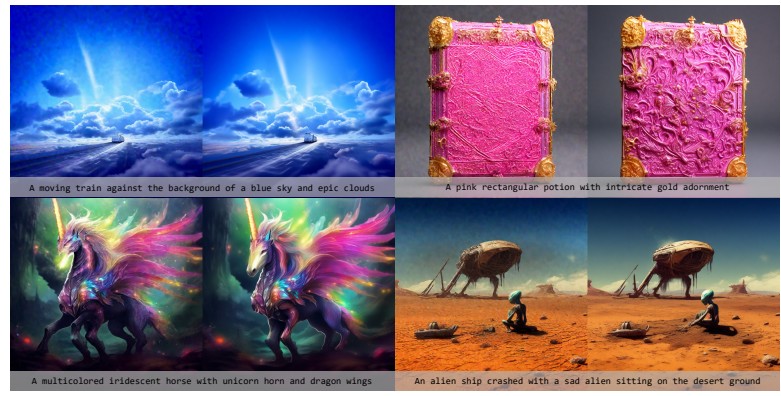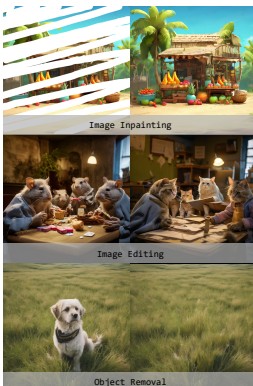

(a) Results of Heun's (2.6 s / img, *left*) and **FlowTurbo** (**1.8 s / img**, *right*)  (b) Extensions

Figure 4: **Qualitative results. (a)** We compared our FlowTurbo with Heun's method on Lumina-Next-T2I [9]. With better image quality, our method requires much less sampling time ($-30.8\%$). **(b)** Since FlowTurbo remains the multi-step sampling paradigm, it can be seamlessly applied to more applications such as image inpainting, image editing, and object removal.

2-order convergence, which requires a 2-order intermediate result as initialization. This explains why we need several Heun's steps at the beginning. To further investigate how the order of the blocks would affect the sampling speed and quality, we perform experiments and summarize the results in Tables 4a and 4b. First, we find changing the order of the sampling blocks will cause worse sampling quality. Second, we show that sequentially using multiple blocks (*e.g.*, $[H_{N_H} P_{N_P} R_{N_R}]_{\times k}$) will cost the same inference time as $H_{kN_H} P_{kN_P} R_{kN_R}$ but lead to worse visual quality.

**Qualitative results and extensions.** We provide qualitative results of high-resolution text-to-image generation by applying FlowTurbo to the newly released flow-based model Lumina-Next-T2I [9]. Since Lumina-Next-T2I adopts a heavy language model Gemma-2B [39] to extract text features and generates high-resolution images ($1024 \times 1024$), the inference speed of it is slower than SiT [24]. In Figure 4a, we show that our FlowTurbo can generate images with better quality and higher inference speed compared with the baseline Heun's method. Besides, since FlowTurbo remains the multi-step sampling paradigm, it can be seamlessly applied to more applications like image inpainting, image editing, and object removal (Section 4.4). Please also refer to the Appendix C for the detailed implementation of various tasks.

**Limitations and broader impact.** Despite the effectiveness of FlowTurbo, our velocity refiner highly relies on the observation that the velocity is a "stable value" during the sampling. However, we have not found such a stable value in diffusion-based models yet, which might limit the application. Besides, the abuse of FlowTurbo may also accelerate the generation of malicious content.

# 5 Conclusion

In this paper, we introduce FlowTurbo, a novel framework designed to accelerate flow-based generative models. By leveraging the stability of the velocity predictor's outputs, we propose a lightweight velocity refiner to adjust the velocity field offsets. This refiner comprises only about 5% of the original velocity predictor's parameters and can be efficiently trained in under 6 GPU hours. Additionally, we have proposed a pseudo corrector that reduces the number of model evaluations while maintaining the same convergence order as the second-order Heun's method. Furthermore, we propose a sample-aware compilation technique to enhance sampling speed. Extensive experiments on various flow-based generative models demonstrate FlowTurbo's effectiveness on both class-conditional image generation and text-to-image generation. We hope our work will inspire future efforts to accelerate flow-based generative models across various application scenarios.

# Acknowledgments

This work was supported in part by the National Natural Science Foundation of China under Grant 62321005, Grant 624B1026, Grant 62336004, and Grant 62125603.

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

# A Detailed Background of Diffusion and Flow-based Models

In this section, we will provide a detailed background of diffusion and flow-based models, which is helpful to understand the difference and relationship between them.

## A.1 Diffusion Models

The *forward pass* i.e. *diffusioin pass* of DPMs can be defined as a sequence of variables $\{\mathbf{x}_t\}_{t\in[0,1]}$ starting with $x_0$, such that for any $t \in [0,1]$, $\mathbf{x}_0 \in \mathbb{R}^D$ is a D-dimensional random variable with an unknown data distribution $p_0(\mathbf{x}_0)$. the distribution of $\mathbf{x}_t$ conditioned on $x_0$ satisfies

$$p_{0t}(\mathbf{x}_t|\mathbf{x}_0) = \mathcal{N}(\mathbf{x}_t|\alpha_t\mathbf{x}_0, \sigma_t\mathbf{I}) \tag{16}$$

where $\alpha_t$, $\sigma_t \in \mathbb{R}^+$ are differentiable functions of $t$ with bounded derevatives. The choice for $\alpha_t$ and $\sigma_t$ is referred to as the noise schedule of a DPM. Let $p_t(x_t)$ denote the marginal distribution of $\mathbf{x}_t$, DPMs choose noise schedules to ensure the marginal distribution $p_1(\mathbf{x}_1) = \mathcal{N}(0, \mathbf{I})$ and the signal-to-noise-ratio (SNR) $\alpha_t^2/\sigma_t^2$ is strictly decreasing w.r.t. $t$ [13]. And we have

$$\mathbf{x}_t = \alpha_t\mathbf{x}_0 + \sigma_t\boldsymbol{\epsilon}, \quad t \in [0,1], \quad \boldsymbol{\epsilon} \sim \mathcal{N}(0, \mathrm{I}) \tag{17}$$

Moreover, Kingma *et al.* [13] prove that the following stochastic differential equation (SDE) has the same transition distribution $q_{0t}(\mathbf{x}_t|\mathbf{x}_0)$ as in (16) for any $t \in [0,1]$:

$$\mathrm{d}\mathbf{x}_t = f(t)\mathbf{x}_t\mathrm{d}t + g(t)\mathrm{d}\mathbf{w}_t, \; t \in [0,1], \quad \mathbf{x}_0 \sim p_0(\mathbf{x}_0) \tag{18}$$

where $\mathbf{w}_t \in \mathbb{R}^D$ is the standard Wiener process , and

$$f(t) = \frac{\mathrm{d}\log\alpha_t}{\mathrm{d}t}, \quad g^2(t) = \frac{\mathrm{d}\sigma_t^2}{\mathrm{d}t} - 2\frac{\mathrm{d}\log\alpha_t}{\mathrm{d}t}\sigma_t^2 \tag{19}$$

Song et al. [38] have shown that the forward process in (18) has an equivalent reverse process from time 1 to 0 under some regularity conditions, starting with the marginal distribution $p_T(\mathbf{x}_T)$:

$$\mathrm{d}\mathbf{x}_t = [f(t)\mathbf{x}_t - g^2(t)\nabla_{\mathbf{x}}\log p_t(\mathbf{x}_t)]\mathrm{d}t + g(t)\mathrm{d}\bar{\mathbf{w}}_t, \quad \mathbf{x}_T \sim p_T(\mathbf{x}_T) \tag{20}$$

where $\bar{\mathbf{w}}_t \in \mathbb{R}^D$ is a standard Wiener process in the reverse time. To solve the reverse process in (20), the only thing we should do is to estimate the score term $\nabla_{\mathbf{x}}\log p_t(\mathbf{x}_t)$ at each time t. In practice, DPMs train a neural network $\boldsymbol{\epsilon}_\theta(\mathbf{x}, t)$ parameterized by $\theta$ to estimate the scaled score function: $-\sigma_t\nabla_{\mathbf{x}}\log p_t(\mathbf{x}_t)$. The parameter $\theta$ is optimized by minimizing the following objective [10, 38, 24]

$$\mathcal{L}_{\mathrm{DM}}(\theta) = \mathbb{E}_{t,p_0(\mathbf{x}_0),p(\mathbf{x}_t|\mathbf{x}_0)} \left[\lambda(t)\|\boldsymbol{\epsilon}_\theta(\mathbf{x}_t, t) + \sigma_t\nabla_{\mathbf{x}}\log p_t(\mathbf{x}_t)\|_2^2\right] \tag{21}$$

where $\lambda(t)$ is a time-dependent coefficient. As $\boldsymbol{\epsilon}_\theta(\mathbf{x}_t, t)$ can alse be regarded as predicting the Gaussian noise added to $\mathbf{x}_t$, it is usually called the *noise prediction model* . Since the ground truth of $\boldsymbol{\epsilon}_\theta(\mathbf{x}_t, t)$ is $-\sigma_t\nabla_{\mathbf{x}}\log p_t(\mathbf{x}_t)$, DPMs replace the score function in (20) by $-\boldsymbol{\epsilon}_\theta(\mathbf{x}_t, t)/\sigma_t$ and we refer to it as *diffusion-based* generative model. DPMs define a parameterized *reverse process* (diffusion SDE) from time 1 to 0, starting with $x_1 \sim p_1(\mathbf{x}_1)$:

$$\mathrm{d}\mathbf{x}_t = \left[f(t)\mathbf{x}_t + \frac{g^2(t)}{\sigma_t}\boldsymbol{\epsilon}_\theta(\mathbf{x}_t, t)\right]\mathrm{d}t + g(t)\mathrm{d}\bar{\mathbf{w}}_t, \quad x_1 \sim \mathcal{N}(0, \mathbf{I}) \tag{22}$$

Samples can be generated from DPMs by solving the diffusion SDE in (22) with numerical solvers.

When discretizing SDEs, the step size is limited by the randomness of the Wiener process. A large step size (small number of steps) often causes non-convergence, especially in high dimensional spaces. For faster sampling, we can consider the associated probability flow ODE [38] which has the same marginla distribution at each time $t$ as that of the SDE. Specifically, for DPMs, Song *et al.* [38] proved that the *probability flow* ODE of (22) is

$$\frac{\mathrm{d}\mathbf{x}_t}{\mathrm{d}t} = \mathbf{v}(\mathbf{x}_t, t) := f(t)\mathbf{x}_t + \frac{g^2(t)}{2\sigma_t}\boldsymbol{\epsilon}_\theta(\mathbf{x}_t, t), \quad \mathbf{x}_1 \sim \mathcal{N}(0, \mathbf{I}) \tag{23}$$

Samples can be generated by solving the ODE from 1 to 0. Comparing with SDEs, ODEs can be solved with larger step sizes as they have no randomness. Furthermore, we can take advantage of efficient numerical ODE solvers to accelerate the sampling.

## A.2 Flow-based Models

To introduce flow in detail, first we construct a *time-dependent* vector field, $u : [0, 1] \times \mathbb{R}^D \to \mathbb{R}^D$. A vector field $u_t$ can be used to construct a time-dependent diffeomorphic map, called a *flow*, $\phi : [0, 1] \times \mathbb{R}^D \to \mathbb{R}^D$, defined via the ordinary differential equation (ODE):

$$\frac{\mathrm{d}}{\mathrm{d}t} \phi_t(\mathbf{x}_0) = u_t(\phi_t(\mathbf{x}_0)) \tag{24}$$

$$\phi_0(\mathbf{x}_0) = \mathbf{x}_0 \tag{25}$$

Chen *et al.* [5] suggested modeling the vector field $u_t$ with a neural network $\mathbf{v}_\theta$, which in turn leads to a deep parametric model of the flow $\phi_t$, called a *Continuous Normalizing Flow* (CNF). It is a more generic modeling technique and can capture the probability paths of diffusion process as well. Training a CNF becomes more practical since the propose of the *conditional flow matching* (CFM) technique [17], which learns the conditional velocity field of the flow.

For generative models, similar to (17) we can add some constraints to the noise schedule such that $\alpha_0 = 1, \sigma_0 = 0$ and $\alpha_1 = 0, \sigma_1 = 1$, and then define the flow as:

$$\psi_t(\cdot|\boldsymbol{\epsilon}) : \mathbf{x}_0 \mapsto \alpha_t \mathbf{x}_0 + \sigma_t \boldsymbol{\epsilon} \tag{26}$$

The corresponding velocity vector field which is used to construct the flow $\psi_t$ can be represented as:

$$u_t(\psi_t(\mathbf{x}_0|\boldsymbol{\epsilon})|\epsilon) = \frac{\mathrm{d}}{\mathrm{d}t} \psi_t(\mathbf{x}_0|\epsilon) = \dot{\alpha}_t \mathbf{x}_0 + \dot{\sigma}_t \boldsymbol{\epsilon} \tag{27}$$

Consider the time-dependent probability density function (PDF) $p_t(\mathbf{x})$ of $\mathbf{x}_t = \psi_t(x_0|\epsilon) = \alpha_t \mathbf{x}_0 + \sigma_t \boldsymbol{\epsilon}$. Lipman *et al.* [17] proved that the marginal vector field $u_t$ that generates the probability path $p_t$ satisfies a Partial Differential Equation (PDE) called *continuity equation* (also *transport equation*)

$$\frac{\mathrm{d}}{\mathrm{d}t} p_t(\mathbf{x}) + \nabla_{\mathbf{x}} \cdot (u_t(\mathbf{x}) p_t(\mathbf{x})) = 0 \tag{28}$$

Using conditional flow matching technique $\mathbf{v}(\mathbf{x}_t, t)$ in (23) can be estimated parametrically as $\mathbf{v}_\theta(\mathbf{x}_t, t)$ by minimizing the following objective

$$\mathcal{L}_{FM}(\theta) = \mathbb{E}_{t, p_1(\boldsymbol{\epsilon}), p_0(x_0)} \left\| \mathbf{v}_\theta(\mathbf{x}_t, t) - \frac{\mathrm{d}}{\mathrm{d}t} \psi_t(\mathbf{x}_0|\epsilon) \right\|_2^2 \tag{29}$$

$$= \mathbb{E}_{t, p_1(\boldsymbol{\epsilon}), p_0(x_0)} \left\| \mathbf{v}_\theta(\mathbf{x}_t, t) - \dot{\alpha}_t \mathbf{x}_0 - \dot{\sigma}_t \boldsymbol{\epsilon} \right\|_2^2 \tag{30}$$

We refer to (23) as a *flow-based* generative model. Since we have $\mathbf{x}_t = \psi_t(x_0|\epsilon)$, the sampling of a flow-based model can be achieved by solving the probability flow ODE with learned velocity

$$\frac{\mathrm{d}\mathbf{x}_t}{\mathrm{d}t} = \mathbf{v}_\theta(x_t, t), \quad x_1 \sim p_1(x_1) \tag{31}$$

## A.3 Relationship Between Diffusion and Flow-based Models

There exists a straightforward connection between $\mathbf{v}(\mathbf{x}_t, t)$ and the score term $\sigma_t \nabla_{\mathbf{x}} \log p_t(\mathbf{x}_t)$ according to [24].

$$\mathbf{v}(\mathbf{x}_t, t) = \frac{\dot{\alpha}_t}{\alpha_t} \mathbf{x}_t + \left( \dot{\sigma}_t - \frac{\dot{\alpha}_t \sigma_t}{\alpha_t} \right) (-\sigma_t \nabla_{\mathbf{x}} \log p_t(\mathbf{x}_t)) \tag{32}$$

---

**Algorithm 1** Heun's Method Sampler

---

**Require:** timesteps $\{t_i\}_{i=0}^{N-1}$, $\alpha_t, \sigma_t, \mathbf{x}_0 \sim \mathcal{N}(0, \mathbf{I})$, velocity prediction model $\mathbf{v}_\theta(\mathbf{x}, t|\mathbf{y})$
**for** $i = 0$ **to** $N - 1$ **do**
    $\Delta t_i \leftarrow t_{i+1} - t_i$
    $\mathbf{d}_i \leftarrow \mathbf{v}_\theta(\mathbf{x}_i, t_i|\mathbf{y})$
    $\tilde{\mathbf{x}}_{t_{i+1}} \leftarrow \mathbf{x}_i + \Delta t_i \mathbf{d}_i$
    $\mathbf{d}_{i+1} \leftarrow \mathbf{v}_\theta(\tilde{\mathbf{x}}_{t_{i+1}}, t_{i+1}|\mathbf{y})$
    $\mathbf{x}_{t_{i+1}} \leftarrow \mathbf{x}_i + \frac{\Delta t_i}{2}[\mathbf{d}_i + \mathbf{d}_{i+1}]$
**end for**
**return:** $\mathbf{x}_N$

---

---

**Algorithm 2** Pseudo Corrector Sampler

---

**Require:** timesteps $\{t_i\}_{i=0}^{N-1}$, $\alpha_t, \sigma_t, \mathbf{x}_0 \sim \mathcal{N}(0, \mathbf{I})$, velocity prediction model $\mathbf{v}_\theta(\mathbf{x}, t|\mathbf{y})$
$\Delta t \leftarrow t_1 - t_0$
**for** $i = 0$ **to** $N - 1$ **do**
    $\Delta t_i \leftarrow t_{i+1} - t_i$
    **if** $i = 0$ **then**
        $\mathbf{d}_i \leftarrow \mathbf{v}_\theta(\mathbf{x}_i, t_i|\mathbf{y})$
    **end if**
    $\tilde{\mathbf{x}}_{t_{i+1}} \leftarrow \mathbf{x}_i + \Delta t_i \mathbf{d}_i$
    $\mathbf{d}_{i+1} \leftarrow \mathbf{v}_\theta(\tilde{\mathbf{x}}_{t_{i+1}}, t_{i+1}|\mathbf{y})$
    $\mathbf{x}_{t_{i+1}} \leftarrow \mathbf{x}_i + \frac{\Delta t_i}{2}[\mathbf{d}_i + \mathbf{d}_{i+1}]$
**end for**
**return:** $\mathbf{x}_N$

---

We can define $\zeta_t = \dot{\sigma}_t - \frac{\dot{\alpha}_t \sigma_t}{\alpha_t}$, and we have $\boldsymbol{\epsilon}_\theta(\mathbf{x}_t, t)$ to estimate $-\sigma_t \nabla_\mathbf{x} \log p_t(\mathbf{x}_t)$, then derive the relationship between $\mathcal{L}_{\mathrm{DM}}(\theta)$ and $\mathcal{L}_{\mathrm{FM}}(\theta)$ We can plug (32) into the loss $\mathcal{L}_{FM}(\theta)$ in Equation (30)

$$\mathcal{L}_{\mathrm{FM}}(\theta) = \mathbb{E}_{t, p_1(\boldsymbol{\epsilon}), p_0(x_0)} \|\mathbf{v}_\theta(\mathbf{x}_t, t) - \dot{\alpha}_t \mathbf{x}_0 - \dot{\sigma}_t \boldsymbol{\epsilon}\|_2^2 \tag{33}$$

$$= \mathbb{E}_{t, p_1(\boldsymbol{\epsilon}), p_0(x_0)} \left\| \frac{\dot{\alpha}_t}{\alpha_t}\mathbf{x}_t + \zeta_t \boldsymbol{\epsilon}_\theta(\mathbf{x}_t, t) - \dot{\alpha}_t \mathbf{x}_0 - \dot{\sigma}_t \boldsymbol{\epsilon} \right\|_2^2 \tag{34}$$

$$= \mathbb{E}_{t, p_1(\boldsymbol{\epsilon}), p_0(x_0)} \left\| \frac{\dot{\alpha}_t \sigma_t}{\alpha_t}\boldsymbol{\epsilon} + \zeta_t \boldsymbol{\epsilon}_\theta(\mathbf{x}_t, t) - \dot{\alpha}_t \mathbf{x}_0 - \dot{\sigma}_t \boldsymbol{\epsilon} \right\|_2^2 \tag{35}$$

$$= \mathbb{E}_{t, p_1(\boldsymbol{\epsilon}), p_0(x_0)} \|\zeta_t \boldsymbol{\epsilon}_\theta(\mathbf{x}_t, t) - \zeta_t \boldsymbol{\epsilon}\|_2^2 \tag{36}$$

$$= \mathbb{E}_{t, p_1(\boldsymbol{\epsilon}), p_0(x_0)} \left[ \zeta_t^2 \|\boldsymbol{\epsilon}_\theta(\mathbf{x}_t, t) - \boldsymbol{\epsilon}\|_2^2 \right] \tag{37}$$

$$\overset{\mathbf{x}_t = \alpha_t \mathbf{x}_0 + \sigma_t \boldsymbol{\epsilon}}{=\!=\!=} \mathbb{E}_{t, p_0(x_0), p(\mathbf{x}_t|\mathbf{x}_0)} \left[ \zeta_t^2 \|\boldsymbol{\epsilon}_\theta(\mathbf{x}_t, t) + \sigma_t \nabla_\mathbf{x} \log p_t(\mathbf{x}_t)\|_2^2 \right] \tag{38}$$

Recall that

$$\mathcal{L}_{\mathrm{DM}}(\theta) = \mathbb{E}_{t, p_0(\mathbf{x}_0), p(\mathbf{x}_t|\mathbf{x}_0)} \left[ \lambda(t) \|\boldsymbol{\epsilon}_\theta(\mathbf{x}_t, t) + \sigma_t \nabla_\mathbf{x} \log p_t(\mathbf{x}_t)\|_2^2 \right], \tag{39}$$

we can see that the difference of $\mathcal{L}_{\mathrm{DM}}(\theta)$ and $\mathcal{L}_{\mathrm{FM}}(\theta)$ during training is caused by the weighted function, which leaving to different trajectories and properties.

## B   Proof of Convergence of Pseudo Corrector

In this section, we will prove that the proposed pseudo corrector has the same local truncation error and global convergence order as Heun's method. The detailed sampling procedure of Heun's method

and pseudo corrector are provided in Algorithm 1 and Algorithm 2. In this section, we use $\mathbf{x}_{t_i}$ to represent the intermediate sampling result at the $t_i$ timestep, and use $\mathbf{x}_{t_i}^* = \mathbf{x}(t_i)$ to denote the corresponding ground-truth value on the trajectory. In all the proofs in this section, we omit the condition $\mathbf{y}$ for simplicity.

## B.1 Assumptions

**Assumption B.1.** *The velocity predictor $\mathbf{v}_\theta(\mathbf{x}, t)$ is Lipschitz continous of constant $L$ w.r.t $\mathbf{x}$.*

**Assumption B.2.** *The velocity predictor $\mathbf{v}_\theta(\mathbf{x}, t)$ has at least 2 derivatives $\frac{\mathrm{d}}{\mathrm{d}t}\mathbf{v}_\theta(\mathbf{x}, t)$ and $\frac{\mathrm{d}^2}{\mathrm{d}t^2}\mathbf{v}_\theta(\mathbf{x}, t)$ and the derivatives are continuous.*

**Assumption B.3.** $h = \max_{0 \le i \le N-1} h_i = \mathcal{O}(1/N)$, *where $N$ is the total number of sampling steps.*

All the above are common in the analysis of the convergence order of fast samplers [22, 23, 42] of diffusion models.

## B.2 Local Convergence

We start by studying the local convergence and Heun's method. Considering the updating from $t_i$ to $t_{i+1}$ and assume all previous results are correct (see the definition of local convergence [15]). The Taylor's expansion of $\mathbf{x}_{t_{i+1}}^*$ at $t_i$ gives:

$$\mathbf{x}_{t_{i+1}}^* = \mathbf{x}_{t_i} + h_i \mathbf{x}^{(1)}(t_i) + \frac{h_i^2}{2}\mathbf{x}^{(2)}(t_i) + \frac{h_i^3}{6}\mathbf{x}^{(3)}(t_i) + \mathcal{O}(h^4). \tag{40}$$

On the other hand, let $\bar{\mathbf{x}}_{t_{i+1}}$ be the prediction assuming $\mathbf{x}_i$ is correct, the updating rule of Heun's method shows:

$$\bar{\mathbf{x}}_{t_{i+1}} = \mathbf{x}_{t_i} + \frac{h_i}{2}[\mathbf{d}_i + \mathbf{d}_{i+1}] \tag{41}$$

$$= \mathbf{x}_{t_i} + \frac{h_i}{2}[\mathbf{x}^{(1)}(t_i) + \mathbf{x}^{(1)}(t_i) + h_i \mathbf{x}^{(2)}(t_i) + \frac{h_i^2}{2}\mathbf{x}^{(3)}(t_i) + \mathcal{O}(h^3)] \tag{42}$$

$$= \mathbf{x}_i + h_i \mathbf{x}^{(1)}(t_i) + \frac{h_i^2}{2}\mathbf{x}^{(2)}(t_i) + \frac{h_i^3}{4}\mathbf{x}^{(3)}(t_i) + \mathcal{O}(h^4) \tag{43}$$

Therefore, the local truncation error can be computed by:

$$T_{i+1} = \|\mathbf{x}_{t_{i+1}}^* - \bar{\mathbf{x}}_{t_{i+1}}\| = \| - \frac{h_i^3}{12}\mathbf{x}^{(3)}(t_i) + \mathcal{O}(h^4)\| \le C_1 h^3, \tag{44}$$

which indicates that Heun's method has 2 order of accuracy.

It is also noted that the local truncation error of the predictor step (which is the same as Euler's method) can be similarly derived by:

$$\tilde{T}_{i+1} = \|\mathbf{x}_{t_{i+1}}^* - \tilde{\mathbf{x}}_{t_{i+1}}\| = \|\frac{h^2}{2}\mathbf{x}^{(2)}(t_i) + \mathcal{O}(h^2)\| \le C_2 h^2. \tag{45}$$

For pseudo corrector, the analysis of local convergence is the same since we need to assume all previous results (including the $\mathbf{x}_{t_i}$ and $\mathbf{d}_i$), which means the local truncation error of pseudo corrector is the same as the Heun's method.

## B.3 Global Convergence

**Global convergence for Heun's method.** When analyzing global convergence, we need to take into account both the local truncation error and the effects of the error of previous results. According to the Lipschitz condition, we have:

$$\|\mathbf{x}_{t_{i+1}}^* - \tilde{\mathbf{x}}_{t_{i+1}}\| \le (1 + hL)\|\mathbf{x}_{t_i}^* - \mathbf{x}_{t_i}\| + C_2 h^2 \tag{46}$$

and

$$\|\mathbf{x}_{t_{i+1}}^* - \mathbf{x}_{t_{i+1}}\| \le (1 + \frac{hL}{2})\|\mathbf{x}_{t_i}^* - \mathbf{x}_{t_i}\| + \frac{hL}{2}\|\mathbf{x}_{t_{i+1}}^* - \tilde{\mathbf{x}}_{t_{i+1}}\| + C_1 h^3. \tag{47}$$

Combining the above two inequalities together, we have

$$\|\mathbf{x}_{t_{i+1}}^* - \mathbf{x}_{t_{i+1}}\| \le (1 + hL + \frac{h^2 L^2}{2})\|\mathbf{x}_{t_i}^* - \mathbf{x}_{t_i}\| + C_3 h^3, \tag{48}$$

where $C_3 = \frac{hLC_2}{2} + C_1$. Note that $\|x_{t_0}^* - x_{t_0}\| = 0$ (their is no error at the beginning of the sampling), it can be easily derived that

$$\|\mathbf{x}_{t_N}^* - \mathbf{x}_{t_N}\| \le \frac{C_3 h^2}{L + \frac{hL^2}{2}}((1 + hL + \frac{h^2 L^2}{2})^N - 1) \le C_4 h^2 (e^{C_5} - 1) = C_6 h^2. \tag{49}$$

Therefore, we have proven that Heun's method have 2 order of global convergence.

**Global convergence for pseudo corrector.** The only difference between pseudo corrector and Heun's method is how $\mathbf{d}_i$ is obtained. Pseudo corrector reuse the $\mathbf{d}_i$ from the last sampling step rather than re-compute it as in Heun's method. As a result, $\mathbf{d}_i$ used in pseudo corrector is computed on $\tilde{\mathbf{x}}_{t_i}$ rather than $\mathbf{x}_{t_i}$, which will lead to another error term when analyzing the global convergence. Concretely, the global error of pseudo corrector can be computed by:

$$\|\mathbf{x}_{t_{i+1}}^* - \tilde{\mathbf{x}}_{t_{i+1}}\| \le (1 + hL)\|\mathbf{x}_{t_i}^* - \mathbf{x}_{t_i}\| + hL\|\tilde{\mathbf{x}}_{t_i} - \mathbf{x}_{t_i}\| + C_2 h^2 \tag{50}$$

$$\|\mathbf{x}_{t_{i+1}}^* - \mathbf{x}_{t_{i+1}}\| \le (1 + \frac{hL}{2})\|\mathbf{x}_{t_i}^* - \tilde{\mathbf{x}}_{t_i}\| + \frac{hL}{2}\|\mathbf{x}_{t_{i+1}}^* - \tilde{\mathbf{x}}_{t_{i+1}}\| + \frac{hL}{2}\|\mathbf{x}_{t_i} - \tilde{\mathbf{x}}_{t_i}\| + C_1 h^3. \tag{51}$$

For the sake of simplicity, let $\tilde{\Delta}_i = \|\mathbf{x}_{t_i}^* - \tilde{\mathbf{x}}_{t_i}\|$ and $\Delta_i = \|\mathbf{x}_{t_i}^* - \mathbf{x}_{t_i}\|$. Therefore, the above formulas becomes:

$$\tilde{\Delta}_{i+1} \le \Delta_i + hL\tilde{\Delta}_i + C_2 h^2 \tag{52}$$

$$\Delta_{i+1} \le (1 + \frac{hL}{2})\Delta_i + \frac{hL}{2}(1 + \frac{hL}{2})\tilde{\Delta}_i + C_4 h^3. \tag{53}$$

By calculating (52)×$hL$+(53) we have:

$$\begin{aligned}
\Delta_{i+1} + hL\tilde{\Delta}_{i+1} &\le (1 + \frac{hL}{2})\Delta_i + \frac{hL}{2}(1 + \frac{hL}{2})\tilde{\Delta}_i + C_4 h^3 + hL\Delta_i + h^2 L^2 \tilde{\Delta}_i + C_2 L h^3 \\
&= (1 + \frac{3}{2}hL)\left[\Delta_i + \frac{\frac{hL}{2} + \frac{5}{4}h^2 L^2}{1 + \frac{3}{2}hL}\tilde{\Delta}_i\right] + C_7 h^3 \\
&\le (1 + \frac{3}{2}hL)(\Delta_i + hL\tilde{\Delta}_i) + C_7 h^3.
\end{aligned} \tag{54}$$

Note that $\Delta_0 + hL\tilde{\Delta}_0 = 0$. Let $\Delta_i' = \Delta_i + hL\tilde{\Delta}_i$, we have

$$\Delta_i' \le (1 + \frac{3}{2}hL)\Delta_i' + C_7 h^3. \tag{55}$$

Similar to the derivation of (49), we can derive that

$$\Delta_i' \le C_8 h^2 ((1 + \frac{3}{2}hL)^N - 1) \le C_9 h^2, \tag{56}$$

which indicates that

$$\Delta_N \le C_9 h^2, \quad hL\tilde{\Delta}_{i+1} \le C_9 h^2. \tag{57}$$

Therefore we have $\Delta_N \le C_9 h^2$, and thus the global convergence of pseudo corrector is 2-order.

Table 6: **Ablation of the number of the velocity refiners.** We change the number of velocity refiners and compare the sampling quality of each configuration. We find there exists a optimal number of velocity refiners to achieve the lowest FID.

| Method | Sample Config | Ratio of Refiner | FID $\downarrow$ | Latency (ms / img) |
|---|---|---|---|---|
| *SiT-XL [8], ImageNet* $(256 \times 256)$ | | | | |
| FlowTurbo | $H_8 P_9 R_6$ | 0.26 | 2.19 | 100.7 |
| FlowTurbo | $H_8 P_9 R_5$ | 0.23 | **2.12** | 100.3 |
| FlowTurbo | $H_8 P_9 R_4$ | 0.19 | 2.18 | 99.9 |
| FlowTurbo | $H_8 P_9 R_3$ | 0.15 | 2.15 | 99.6 |
| FlowTurbo | $H_6 P_9 R_6$ | 0.29 | 2.25 | 87.2 |
| FlowTurbo | $H_6 P_9 R_5$ | 0.25 | **2.20** | 86.8 |
| FlowTurbo | $H_6 P_9 R_4$ | 0.21 | 2.21 | 86.4 |
| FlowTurbo | $H_6 P_9 R_3$ | 0.17 | 2.22 | 86.0 |

## C  Implementation Details

**Class-conditional image generation.**  We use the SiT-XL-2[24] as our base model to perform the experiments on class-conditional image generation. We use a single block of SiT-XL-2 as the Velocity Refiner. We double the input channel from 4 to 8 to take the previous velocity as input. The resulting velocity refiner only contains 29M parameters, about 4.3% of the original SiT-XL-2(675M). We use ImageNet-1K [6][2] to train our velocity model. We used AdamW [21] optimizer for all models. We use a constant learning rate of $5 \times 10^{-5}$ and a batch size of 18 on a single A800 GPU. We used a random horizontal flip with a probability of 0.5 in data augmentation. We did not tune the learning rates, decay/warm-up schedules, AdamW parameters, or use any extra data augmentation during training. Our velocity refiner (for SiT-XL-2) trains at approximately 4.44 steps/sec on an A800 GPU, and converges in 30,000 steps, which takes about 2 hours.

**Text-to-image generation.**  We use the 2-RF in InstaFlow [20] as our base model to perform the experiments on text-to-image generation. Since the architecture of the original velocity predictor in [20] is a U-Net [30], we cannot directly use a single block of it as the velocity refiner as we do for SiT [24]. Instead, we simply reduce the number of channels in each block from [320, 640, 1280, 1280] to [160, 160, 320, 320] and reduce the number of layers in each block from 2 to 1. We also double the input channel from 4 to 8 to take the previous velocity as input. The resulting velocity refiner only contains 43.5M parameters, about 5% of the original U-Net (860M). We use a subset of LAION [34][3] containing only 50K images to train our velocity model. We use AdamW [21] optimizer with a learning rate of 2e-5 and weight decay of 0.0. We adopt a batch size of 16 and set the warming-up steps as 100. We also use a gradient clipping of 0.01 to stabilize training. We train our model on a single A800 GPU for 10K iterations, which takes about 5.5 hours.

**Implementation of extension tasks.**  We have demonstrated our FlowTurbo is also suitable for extension tasks due to the multi-step nature of our framework in Section 4.4. For image inpainting, we adopt the inpainting pipeline in diffusion models [4], where we merge the noise latent and the generated latent at a specific timestep by the input mask. For object removal, we first use a Grounded-SAM [5] to generate the mask and perform similar image inpainting pipeline. For image editing, we adopt the SDEdit [25] which first adds noise to the original image and use it as an intermediate result to continue the sampling.

## D  More Analysis

In this section, we provide more analysis through both quantitative results and qualitative results.

---

[2]License: Custom (research, non-commercial)

[3]License: Creative Common CC-BY 4.0

[4]https://huggingface.co/docs/diffusers/en/using-diffusers/inpaint

[5]https://github.com/IDEA-Research/Grounded-Segment-Anything

Table 7: **Comparisons with state-of-the-art methods on text-to-image generation.** We compare our FlowTurbo with state-of-the-art diffusion models (15 steps DPM-Solver++ [23]) and show our FlowTurbo enjoys favorable trade-offs between sampling quality and speed.

| Method | Sample Config | FLOPs (G) | Latency (ms / img) | FID$\downarrow$ |
|---|---|---|---|---|
| SD2.1 [30] | 15 steps DPM++ [23] | 11427 | 286.0 | 33.03 |
| SDXL [29] | 15 steps DPM++ [23] | 24266 | 427.2 | 29.46 |
| PixArt-$\alpha$ [4] | 15 steps DPM++ [23] | 17523 | 366.8 | 37.96 |
| PixArt-$\sigma$ [3] | 15 steps DPM++ [23] | 17957 | 365.9 | 33.62 |
| FlowTurbo | $H_1 P_6 R_3$ | 4030 | 104.8 | 28.60 |
| FlowTurbo | $H_3 P_6 R_3$ | 5386 | 137.0 | 27.60 |

### D.1 More Quantitative Results

**Ablation of the number of the velocity refiners.** In Table 6, we investigate how to choose the number of velocity refiners to get a better sampling quality. We adopt two basic configurations of $H_7 R_{10}$ and $H_5 R_{10}$, and vary the number of velocity refiners from 3 to 6. We find that the FID will first decrease and then increase when $N_R$ becomes larger, and there exists an optimal $N_R = 5$ where we reach the lowest FID. These results indicate that we can always tune this hyper-parameter to expect a better result.

**More comparisons on text-to-image generation.** In Table Table 7, we compare the sampling quality and speed of FLowTurbo with state-of-the-art diffusion models on text-to-image generation. For all the diffusion models, we adopt a 15-step DPM-Solver++ [23] as the default sampler. The FLOPs reported also take the multi-step sampling into account. Our results show that our FlowTurbo can achieve the lowest FID and inference latency.

### D.2 More Qualitative Results

To better illustrate the sampling quality of our FlowTurbo, we provide more qualitative results on both class-conditional image generation and text-to-image generation.

**Class-conditional image generation.** We use SiT-XL [24] as our flow-based model for class-conditional image generation. In Figure 5, we provide random samples from FlowTurbo of the sample config $H_8 P_9 R_5$, which inference at 100 ms/img. We also demonstrate the sampling quality trade-offs in Figure 6, we compare the sampling quality of two different configurations $H_1 P_5 R_3$ (38 ms / img) and $H_8 P_9 R_5$ (100 ms / img). We generate the images from the same initial noise for better comparisons. Our result demonstrates that our FlowTurbo can achieve real-time image generation, and the sampling quality can be further improved with more computational budgets.

**Text-to-image generation.** We adopt Lumina-Next-T2I [9] to achieve text-to-image generation. We compare the sampling quality and speed of Heun's method and our FlowTurbo in Figure 7. We find that FlowTurbo can consistently generate images with better quality and more visual details, while requiring less inference time.

## E Code

Our code is implemented in PyTorch [6]. We use the codebase of [24] to conduct experiments. The code is available at `https://github.com/shiml20/FlowTurbo`.

---

[6]https://pytorch.org

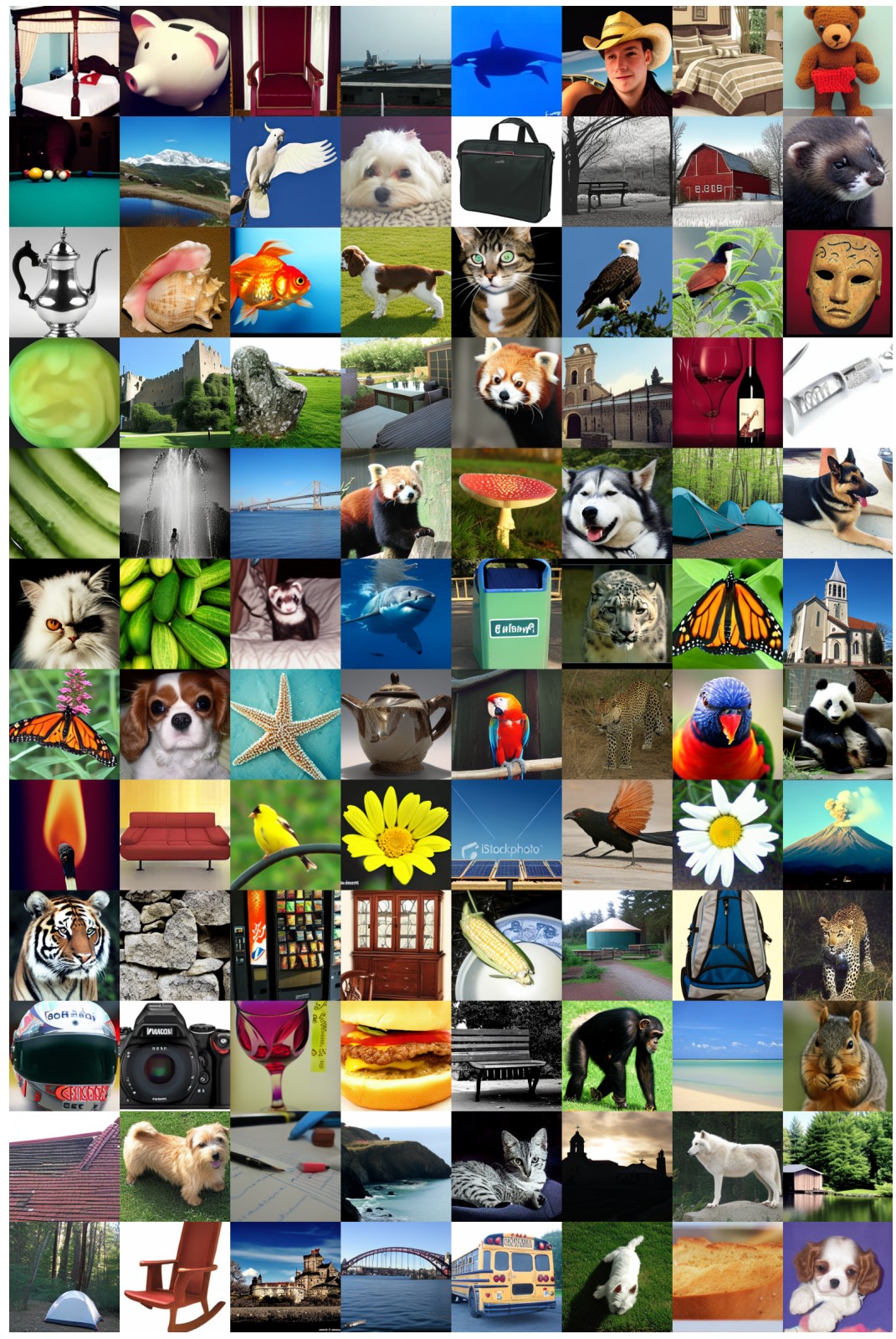

Figure 5: Random samples from **FlowTurbo** on ImageNet 256 × 256. We use a classifier-free guidance scale of 4.0 and the sample config of $H_8 P_9 R_5$ (**100 ms / img**)

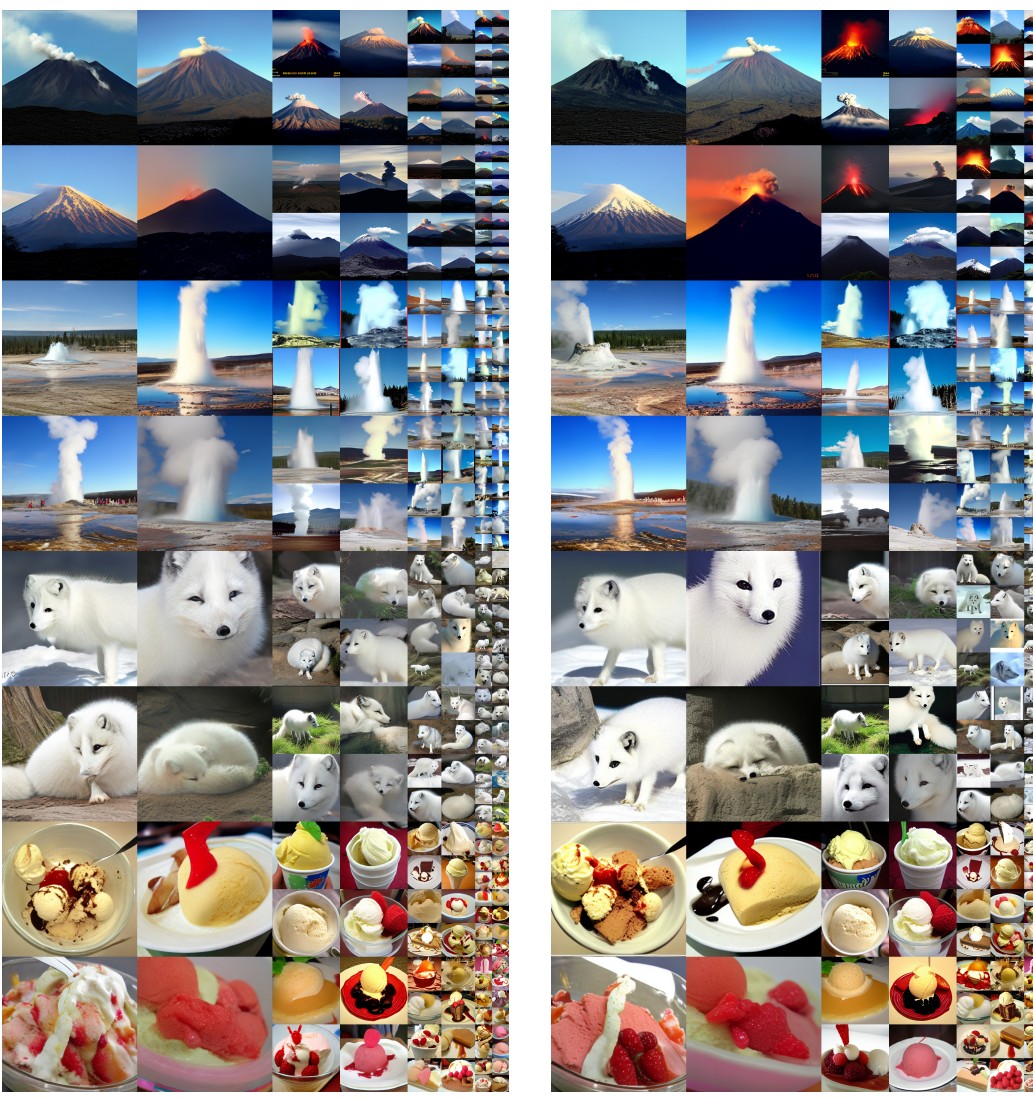

(a) Sample Config $H_1P_5R_3$ (**38 ms / img**)  (b) Sample Config $H_8P_9R_5$ (**100 ms / img**)

Figure 6: Uncurated 256×256 samples from **FlowTurbo** (CFG = 4.0). For better visualization. We compare two sample configurations ($H_1P_5R_3$ and $H_8P_9R_5$). The same initial noise is used for both sample configurations for better comparisons.

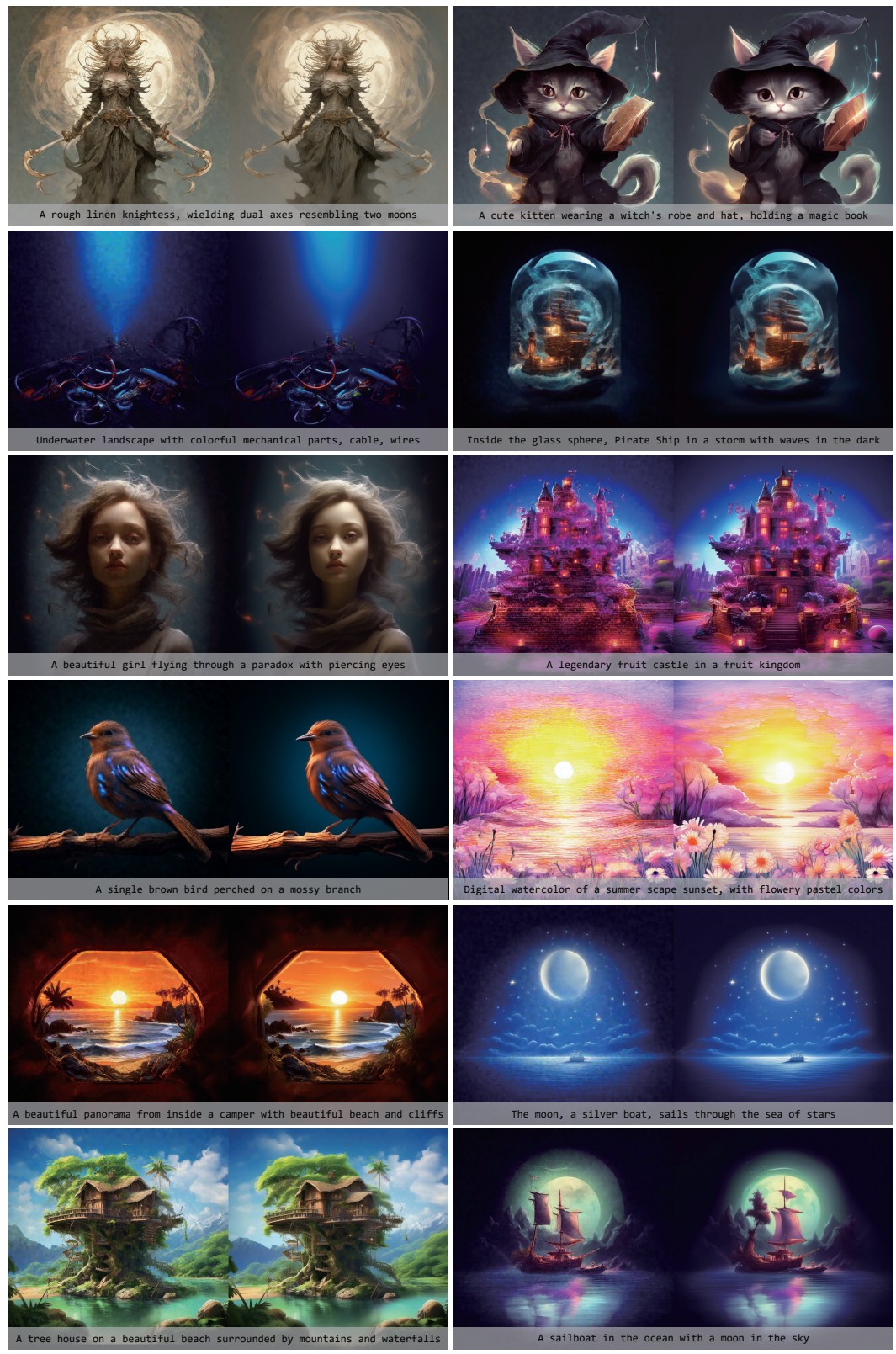

Figure 7: More visual comparisons between Heun's method (2.6 s / img, *left*) and our FlowTurbo (1.8 s / img, *right*).

