# OpenReview forum: "FlowTurbo: Towards Real-time Flow-Based Image Generation with Velocity Refiner"
_NeurIPS.cc/2024/Conference — NeurIPS 2024 poster_

### Official Review · Reviewer_Q4Pt · 2024-07-01

**Soundness:** 3
**Presentation:** 3
**Contribution:** 3
**Rating:** 7
**Confidence:** 4

**Summary:**

This paper introduces FlowTurbo, a framework to accelerate flow-based generative models. The key contributions are:
1) a lightweight velocity refiner to estimate the offset of velocity efficiently during sampling;
2) a pseudo corrector to reduce the number of model evaluations while keeping the original second-order convergence;
3) sample aware compilation to compile model evaluations, sampling steps and cfg into a static graph for extra speedup;

Extensive experiments with both class-conditioned and text-to-image generation show significant acceleration (30-58%). Notably, on ImageNet 256x256 generations, FlowTurbo achieves the fastest sampling speed (38 ms/image) with FID 3.93. Moreover, due to the compatibility with multi-step sampling, it can also be applied to other tasks including image editing, inpainting, etc.

**Strengths:**

Given the popularity of flow-based generation models, SD3 for image and Sora for video, flow-based model acceleration is an important area, which is under-explored compared with diffusion-based models.

Motivated by the stability of velocity prediction during sampling, the authors proposed a lightweight velocity refiner. The pseudo corrector technique reduces the model evaluation while preserving the second-order convergency.  These ideas are novel and effective. The experimental results are comprehensive covering generation task with class-conditional and text-to-image generation, editing tasks including editing, inpainting, object removal. The ablation studies are thorough, examining each component's contribution and various hyperparameters. Finally, the code is also provided for reproducibility. The paper is easy to follow, for example, Figure 2 illustrates the motivation for light-weight estimation model very clearly.

**Weaknesses:**

For text-to-image generation, insta-flow is adopted as the teacher model, which is built on top of sd1.5. It would be great to experiment with other flow-based models that are natively trained with flow matching, for example, sd3.
It would also be interesting to see how FlowTurbo performs when the model size changes. For example, SiT-XL vs SiT-L.
Finally, one missing ablation is the architecture of refiner. how to design a refiner for different architectures?

**Questions:**

Can this method also be applied to video-based generation models? The inference is much slower for video generation compared with image generation. it would be great to see if there is a unified method for both image and video generation.

**Limitations:**

The proposed method is limited to flow-based generation models.

---

> ### Author Rebuttal · Authors · 2024-08-06
>
> We sincerely thank the reviewer for the positive comments on our work! We address the questions and clarify the issues accordingly as described below.
>
> **Q1: About experiments on other models.**
>
> **[Reply]** Thanks for your advice. Since SD3 is open-sourced a few months later than the submission, we didn't include experimental results on it in our original paper. In the following table, we compare the sampling quality (measured by FID 30K $\downarrow$ on MSCOCO) and the sampling speed of the default sampler of SD3 (Euler's method) and our FlowTurbo:
>
> | Method    | FID 30K $\downarrow$ | Latency (s / img) |
> | --------- | -------------------- | ----------------- |
> | Euler     | 28.4                 | 2.25              |
> | FlowTurbo | 28.3                 | 1.35              |
>
> Besides, we also provide qualitative comparisons on SD3 and the newly released SOTA model FLUX.1-dev in Figure A2 and Figure A3 in the attached one-page PDF, where we demonstrate FlowTurbo can generate more details than the baseline.
>
> **Q2: About different model sizes.**
>
> **[Reply]** Thanks for pointing out this. In our original paper, we conducted experiments with models of various sizes including SiT (675M), InstaFlow (860M), Lumina-Next-T2I (2B). We also agree that it is interesting to see the performance on similar architectures with different model sizes such as SiT-XL and SiT-L. However, SiT only released their SiT-XL-2 model and we failed to find another publicly available family of models that satisfied the requirement. We will add the comparison once the SiT releases another model such as SiT-L or SiT-B.
>
> **Q3. About the refiner architecture**
>
> **[Reply]** When designing the architecture for the velocity refiner, we followed a simple rule to make the refiner have a similar architecture as the base velocity predictor but with much fewer parameters (~5% of the base model). The detailed architecture is described in Section 4.1 and Appendix C. For example, since SiT consists of multiple transformer blocks, we simply use a single block as the refiner. For text-to-image generation, we reduce the number of layers and channels of the UNet. In our early experiments, we did have tried another architecture for class-conditional image generation, where a SiT-S (a smaller version of the base velocity predictor SiT-XL) is adopted as the refiner (as shown in the following table):
>
> | Sample Config | Refiner Architecture | Params | FID  |
> | ------------- | -------------------- | ------ | ---- |
> | $H_5 P_7 R_3$ | SiT-S                | 33M    | 2.53 |
> | $H_5 P_7 R_3$ | a block of SiT-XL    | 29M    | 2.22 |
> | $H_8 P_9 R_5$ | SiT-S                | 33M    | 2.24 |
> | $H_8 P_9 R_5$ | a block of SiT-XL    | 29M    | 2.12 |
>
> We find that using a block of SiT-XL as the refiner is slightly better than the SiT-S. These results demonstrate that our framework is robust to the choice of model architectures for the velocity refiner. We will add these discussions and results to Appendix C of the revised paper.
>
>
> **Q4. About the application to video generation models**
>
> **[Reply]** Thanks for your advice. Since our FlowTurbo is designed to accelerate flow-based generative models, it can be applied to video generation models as long as they are trained via flow-matching (i.e., has a stable velocity during the sampling). Specifically, we experiment with the recently released model Open-Sora and provide the qualitative results in Figure A4 of the attached one-page PDF, where we show our FlowTurbo generates video with higher fidelity under the same computational budgets.

---

> > ### Comment · Reviewer_Q4Pt · 2024-08-12
> >
> > Thanks the authors for providing the detailed rebuttal and response to my questions. I will maintain my score.

---

> > > ### Author Response · Authors · 2024-08-13
> > > **Thanks for your time and postive feedback**
> > >
> > > We greatly appreciate your response and valuable suggestions, which will improve the quality and impact of our paper. We will incorporate your feedback and update our paper accordingly in the revision.

---

### Official Review · Reviewer_RYd7 · 2024-07-12

**Soundness:** 3
**Presentation:** 3
**Contribution:** 3
**Rating:** 6
**Confidence:** 4

**Summary:**

This paper explores enhancing flow-based generative models for image generation by accelerating the sampling process while maintaining or improving image quality. The key contribution, FlowTurbo, is a new framework that introduces a lightweight velocity refiner to adjust velocity predictions during sampling, reducing computational costs. Additionally, techniques like pseudo correctors and sample-aware compilation further optimize the sampling speed. The results show significant acceleration in image generation tasks with improved FID scores, establishing a new benchmark for real-time image generation.

**Strengths:**

Overall I find that the writing is clear, concise, and well-structured, making it easy for readers to follow the arguments and understand the key points. This paper starts from the distinct features of flow models and naturally proposes corresponding solutions. Comprehensive ablations validate the effectiveness of the proposed methods.

**Weaknesses:**

- The design of three types of blocks in sampling is somewhat confusing to me. Why do you have to use these types of blocks and arrange them in this order? Do you have to empirically decide the order and numbers of these blocks for each task/domain/model?
- The visualization in Figure 1 is interesting. However, visualizing the discretization error or curvature [1] during sampling seems to be a more suitable and accurate method.
- Some important related works [1,2], tailored for flow models, should be discussed and compared.

[1] Nguyen, Bao, Binh Nguyen, and Viet Anh Nguyen. "Bellman optimal step-size straightening of flow-matching models." *arXiv preprint arXiv:2312.16414* (2023).

[2] Shaul, Neta, et al. "Bespoke Non-Stationary Solvers for Fast Sampling of Diffusion and Flow Models." *arXiv preprint arXiv:2403.01329* (2024).

**Questions:**

If I understand correctly, the proposed method can be applied to diffusion models as well. (Although Figure 1 illustrates flow models may be more suitable.) Have you tested the results on diffusion models like SD or PixArt?

**Limitations:**

Yes.

---

> ### Author Rebuttal · Authors · 2024-08-06
>
> We sincerely thank the reviewer for the positive comments on our work! We address the questions and clarify the issues accordingly as described below.
>
> **Q1: About the design of sampling blocks.**
>
> **[Reply]** Sorry for the confusion. As illustrated in Figure 1, the velocity during the sampling process stabilizes in the final few steps. At this stage, we employ a lightweight refiner to adjust the velocity offset. Additionally, our pseudo corrector is crafted to efficiently achieve second-order convergence, which requires second-order intermediate results (achieved by several Heun's steps at the start) for initialization. Consequently, the sequence of the three block types should be: Heun's block, pseudo corrector block, and refiner block. As the results shown below, changing the order of the blocks will lead to worse sampling quality:
>
> |Method|FID $\downarrow$|Latency (ms / img)|
> |-----|----|-----|
> |$H_5 P_7R_3$|2.24|72.4|
> |$P_5 H_7R_3$|2.38|79.2|
> |$P_5 R_7H_3$|13.66|53.7|
> |$H_5 R_7P_3$|30.94|60.4|
>
> Regarding the number of steps for each block, we have included comparisons in Table 3 and Table 4. Generally, we apply the refiner during the last 20-40% of the sampling steps, where the velocity is relatively stable. We can always adjust the ratio between Heun's steps and pseudo-corrector steps to balance sampling quality and complexity (as seen in Table 3d). Moreover, since reconfiguring the steps doesn't require additional training, we can select an optimal configuration based on computational resources and qualitative results. We will incorporate this detailed analysis and discussion into our revised paper to offer better guidance on designing the configuration of our FlowTurbo.
>
>
> **Q2: About the visualization.**
>
> **[Reply]** Many thanks for your valuable advice. We agree that visualizing the curvature is more suitable and accurate. We provide the comparisons of the curvatures of different models including DiT, SiT, SD3, FLUX.1-dev (a newly released SOTA flow model), Open-Sora (a flow-based video generation model) in the Figure A1 in the attached PDF. The visualization of curvature basically aligns with our original PCA visualization, where we find the flow-based models enjoys much smaller curvature, indicating the sampling trajectory is straighter. We will replace the visualization in Figure 1 with curvature in our revised paper.
>
> **Q3: About the related work.**
>
> **[Reply]** Thanks for pointing out this. We agree that these two papers are related to our paper and we will discuss them in the related work (L107) as follows:
>
> > There are also some works to accelerate the sampling of flow-based models, such as computing the optimal sampling stepsizes [1] and solver distillation approaches [2].
>
>
> In our FlowTurbo, we do not consider the variation of stepsizes during the sampling, which makes our method orthogonal to [1]. In other words, we can combine [1] and our FlowTurbo to improve the sampling efficiency. Following [1], we perform experiments on CelebA-HQ and the results are as follows:
>
> |Method|NFE|FID$\downarrow$|
> |--|---|---|
> |Euler|6|127.01|
> |Bellman [1]|6| 72.54|
> |Bellman [1] + FlowTurbo (ours)|6|**68.74** |
> |Euler|8|109.42|
> |Bellman [1]|8| 49.80|
> |Bellman [1] + FlowTurbo (ours)|8|**45.95** |
>
> In the following table, we compare our FlowTurbo with Bespoke Non-Stationary (BNS) Solvers [2] on ImageNet. Note that BNS only provides the results on ImageNet 64x64 while we adopt the experimental while our experiment setting follows SiT (ImageNet 256x256) and we cannot find a good publicly available flow-based model trained on ImageNet 64x64. The different resolution and base models choices might slightly affect the results. We also report the number of function evaluations (NFE) in the table, where we use $f_1$ and $f_2$ ($f_2\approx 0.1 f_1$) to represent the velocity predictor and refiner, respectively.
>
> |Method|Resolution|ImageNet FID 50K| NFE |Training Costs|
> |-----|-------|-----|-----|------|
> |BNS| 64x64| 2.62|16 | 2-10 GPU days|
> |FlowTurbo ($H_5 P_5 R_2$)| 256x256 |2.46 | 15 $f_1$ + 2 $f_2$ | 2 GPU hours|
>
> We also provide the result on MS COCO text-to-image generation:
> |Method|Resolution|COCO FID 30K| NFE |Training Costs|
> |-----|-------|-----|-----|-----|
> |BNS| 512x512|14.68| 20 | 15-24 GPU days|
> |FlowTurbo ($H_3 P_6 R_3$)| 512x512|11.95 | 12 $f_1$ + 3 $f_2$ | 5.5 GPU hours|
>
> These results clearly show that our FlowTurbo is better in both sampling quality and training efficiency. Besides, BNS solvers struggle to generalize unseen NFEs while our FlowTurbo allows flexible adjustment of the sampling configuration after the velocity has been trained. We will add the above discussion and comparisons in our revised paper.
>
> **Q4: Applying FlowTurbo for diffusion models.**
>
> **[Reply]** That's a good question. In our early experiments, we did try to transfer a similar idea of FlowTurbo to diffusion models. Specifically, we adopted PixArt as the base model (diffusion transformer) and tried to learn a refiner for the $\epsilon_\theta$. However, just as Figure 1 suggests, the $\epsilon_\theta$ of diffusion models changes a lot during the sampling and it is really hard to regress the offset of $\epsilon_\theta$. We have also tried to convert the $\epsilon_\theta$ to an equivalent velocity through Equation (32):
>
> $$\boldsymbol{v}=\frac{\dot{\alpha}_t}{\alpha_t}x + (\dot{\sigma}_t-\frac{\dot{\alpha}_t\sigma_t}{\alpha_t})\boldsymbol{\epsilon},$$
> but the refiner still cannot converge. We think some techniques such like [3] to transform the sampling path to be straighter could be helpful, and we will leave this for future work.
>
> [3] Shaul et. al, Bespoke Solvers for Generative Flow Models, arxiv 2023, ICLR 2024

---

> > ### Comment · Reviewer_RYd7 · 2024-08-10
> >
> > Thanks for this detailed response from the authors. This rebuttal indeed answers many of my questions. Although I still think introducing three types of blocks in a specific order is not the best solution for speeding up, the authors demonstrate its potential, such as in combination with other advanced methods or applying to more advanced models. So, I will raise my score acoordingly.

---

> > > ### Author Response · Authors · 2024-08-10
> > > **Thanks for raising your score and providing valuable feedback**
> > >
> > > Thanks for raising your score and providing valuable feedback. We are glad to know that most of your concerns have been resolved. Regarding the design of our method such as the types of sampling blocks, we will provide more detailed discussions and analyses through both visualizations and experiments (as we have shown in the rebuttal) in the revised paper.

---

### Official Review · Reviewer_PyDm · 2024-07-12

**Soundness:** 3
**Presentation:** 3
**Contribution:** 3
**Rating:** 7
**Confidence:** 5

**Summary:**

The authors propose FlowTurbo, a method to adapt pre-trained flow-based generative models for faster sampling. The method is based on the observation that the predicted velocity field is quite stable throughout the integration time and hence at each timestamp only small refinement from the previous step is required. Therefore, the authors propose to train a lightweight refinement network and use it at certain timestamps instead of the original velocity field predictor. In addition to the refinement network the paper also suggests to speed up the Heun's integration method by reusing the velocities predicted at previous corrector steps. All the sampling blocks are compiled for further speed up. As a result, FlowTurbo demonstrates great trade-off between speed and quality compared to prior work and achieves sota results on certain tasks.

**Strengths:**

The problem of speeding up generative models is of high importance nowadays. While the majority of the methods aims for decreasing the number of denoising steps, the authors of FlowTurbo propose to make each step significantly cheaper. This is achieved at low additional training costs and could have high impact on the methods that explicitly utilise the iterative nature of modern diffusion- or flow-based generative models (such as [1, 2, 3], see references in Weaknesses). The method is well-grounded and carefully ablated. The experiments demonstrate the effectiveness of FlowTurbo compared to default sampling.

**Weaknesses:**

1) **Prior work**: The paper does a good job with contextualizing it relative to the diffusion-based models. However, references to many flow-based methods are missing, which include:

    - Models that leverage flow matching for generative process [2, 4, 5, 6, 7, 8, 9].

    - And (which is more connected to the topic of the paper) the methods that try to speed up the sampling for flow-based models [10, 11, 12, 13].

2) **Method**: Despite the good results, it seems that it is not trivial how to choose the exact configuration of the sampling blocks, i.e. the order and the number of different steps. From the first sight the configurations presented in the paper are quite random. At least some intuition on how to setup those is required. Otherwise, if the configuration needs to be tuned for every other dataset, the method is not that straight-forward to apply.

3) **Evaluation**:
    - The tables would be simpler to read if NFEs (number of function evaluations) were also included as a separate column.
    - Is the step of the baseline method also compiled in tables 1a and 1b? If not, it would be nice to add this experiment, in addition to the ablations, to see the effect of other parts.

[1] Watson et. al, Novel view synthesis with diffusion models, ICLR 2023

[2] Davtyan et. al, Efficient Video Prediction via Sparsely Conditioned Flow Matching, ICCV 2023

[4] Hu et. al, Motion Flow Matching for Human Motion Synthesis and Editing, arxiv 2023

[5] Fischer et. al, Boosting Latent Diffusion with Flow Matching, arxiv 2023

[6] Davtyan et. al, Learn the Force We Can: Enabling Sparse Motion Control in Multi-Object Video Generation, AAAI 2024

[7] Davtyan et. al, Enabling Visual Composition and Animation in Unsupervised Video Generation, arxiv 2024

[8] Gui et. al, Depthfm: Fast monocular depth estimation with flow matching, arxiv 2024

[9] Hu et. al, Flow Matching for Conditional Text Generation in a Few Sampling Steps, arxiv 2024

[10] Shaul et. al, Bespoke Solvers for Generative Flow Models, arxiv 2023, ICLR 2024

[11] Lee et. al, Minimizing Trajectory Curvature of ODE-based Generative Models, ICML 2023

[12] Pooladian et. al, Multisample flow matching: Straightening flows with minibatch couplings, ICML 2023.

[13] Tong et. al, Improving and generalizing flow-based generative models with minibatch optimal transport, TMLR 2023.

[14] Lipman et. al, Flow matching for generative modeling, ICLR 2023.

**Questions:**

1) In the paper the order of blocks $H_{N_H}P_{N_P}R_{N_R}$ is fixed, and only the number of the blocks $N_H$, $N_P$ and $N_R$ changes. What is the reason behind this? Have you tried other schemes, e.g. sequentially using multiple blocks of type $H_{N_H}P_{N_P}R_{N_R}$?

2) A rather nitpicky comment: Multiple times throughout the paper it is mentioned that the linear interpolation corresponds to the optimal transport from the noise distribution to the data distribution. It is not precisely correct, as the linear interpolation corresponds to the optimal transport from the noise distribution to the distribution concentrated around a particular data point [14]. Methods that attempt to steer flow-matching towards optimal transport between noise and data involve additional tricks [12, 13].

Some typos:
1) Line 137: "interpolation between $x_0$ and $\cancel{v}$ $\epsilon$".
2) Line 170: "simulate the $\cancel{x_t}$  $x_{t_i}$".
3) Eq. 15: $\cancel{x_i \leftarrow x_{i-1}}$ $x_{t_i} \leftarrow x_{t_{i - 1}}$

**Limitations:**

The authors have adequately addressed the limitations and the potential negative societal impact of their work.

---

> ### Author Rebuttal · Authors · 2024-08-06
>
> We sincerely thank the reviewer for the positive comments on our work! We address the questions and clarify the issues accordingly as described below.
>
> **Q1: About prior works on flow-based methods.**
>
> **[Reply]** Thanks for your suggestions. We agree that these works are important in the area of flow-based models, and we will include the mentioned papers in the revised paper. Specifically, we will modify L31 to:
>
>
> > Alongside the research on diffusion models, flow-based models have garnered increasing attention due to their versatility in modeling data distributions, and have been widely applied in various domains including image generation [5], video generation [6, 7], video prediction [2], human motion synthesis [4], depth estimation [8], text generation [9], etc.
>
> And for the acceleration methods [10-13] for flow-based models, we will add the discussion of these methods in our related work (L107):
>
> > There are also some work aim to obtain better flow paths which are straighter and more effective for sampling. These work includes some useful technique such as transformed sampling paths [10], minimizing the trajectory curvature [11], multisample flow matching [12], and minibatch optimal transport [13].
>
> **Q2: About the configuration of the sampling blocks.**
>
> **[Reply]** Sorry for the confusion. According to the observation in Figure 1, the velocity during the sampling would become stable at the final few steps, where we adopted a lightweight refiner to regress the velocity offset. Besides, our pseudo corrector is designed to efficiently achieve 2-order convergence, which requires a 2-order intermediate result as initialization. This explains why we need several Heun's steps at the beginning. We also provide some comparisons of different orders and schemes (including repeating the blocks) as follows:
>
> |Method|FID $\downarrow$|Latency (ms / img)|
> |-----|----|-----|
> |$H_5 P_7R_3$|2.24|72.4|
> |$P_5 H_7R_3$|2.38|79.2|
> |$P_5 R_7H_3$|13.66|53.7|
> |$H_5 R_7P_3$|30.94|60.4|
>
> |Method|FID $\downarrow$|Latency (ms / img)|
> |-----|----|-----|
> |$[H_1 P_1R_1]_{\times 2}$| 8.15 | 34.8 |
> |$H_2P_2R_2$|6.39| 34.8 |
> |$[H_1 P_1R_1]_{\times 3}$| 4.34 | 45.4 |
> |$H_3P_3R_3$|3.34| 45.4 |
>
> As for the number of different steps, we have included some comparisons in Table 3 and Table 4. Generally speaking, we choose to apply the refiner at the last $20\sim 40$% sampling steps where the velocity is relatively stable, and we can always adjust the ratio between Heun's steps and pseudo-corrector steps to control the trade-offs between sampling quality and complexity (Table 3d). Besides, since adjusting the configuration does not require further training, we can always select a good configuration according to the computational budgets and qualitative results. We will add these detailed analyses and discussion to our revised paper to provide better guidelines on how to design the configuration of our FlowTurbo.
>
> **Q3: About the evaluation.**
>
> **[Reply]** Thanks for the comments. We provide the results in Table 1a and 1b with a column of NFEs and the baseline methods compiled as follows. We use $f_1$ to denote the function evaluation of the original velocity predictor and $f_2$ to denote the evaluation of our velocity refiner.
>
> **_Table 1a: Class-conditional Image Generation, ImageNet (256x256)_**
> | Method        | Sample Config    | FLOPs (G) | FID 50K $\downarrow$ | Latency (ms / img)   | NFE  |
> |---------------|------------------|-----------|------|-----------------------|------|
> | Heun's        | $H_8$            | 1898      | 3.68 | 68.0                  |  16 $f_1$  |
> | FlowTurbo     | $H_2 P_4 R_2$    | 957       | 3.63 | 41.6       |     8 $f_1$ + 2 $f_2$ |
> | Heun's        | $H_{11}$         | 2610      | 2.79 | 88.2                 |   22 $f_1$ |
> | FlowTurbo     | $H_2 P_8 R_2$    | 1431      | 2.69 | 55.2          |  12 $f_1$ + 2 $f_2$    |
> | Heun's        | $H_{15}$         | 3559      | 2.42 | 115.3                 | 30 $f_1$    |
> | FlowTurbo     | $H_5 P_7 R_3$    | 2274      | 2.22 | 72.5        |   17 $f_1$ + 3 $f_2$   |
> | Heun's        | $H_{24}$         | 5694      | 2.20 | 176.2                 | 48 $f_1$    |
> | FlowTurbo     | $H_8 P_9 R_5$    | 3457      | 2.12 | 100.3       |   25 $f_1$ + 5 $f_2$   |
>
> **_Table 1b: Text-to-image Generation, MS COCO 2017 (512x512)_**
> | Method        | Sample Config    | FLOPs (G) | FID 5K $\downarrow$ | Latency (ms / img)   | NFE  |
> |---------------|------------------|-----------|------|-----------------------|------|
> | Heun's        | $H_4$            | 3955      | 32.77 | 92.0                |   8 $f_1$   |
> | FlowTurbo     | $H_1 P_2 R_2$    | 2649      | 32.48 | 68.4        |   4 $f_1$ + 2 $f_2$    |
> | Heun's        | $H_5$            | 4633      | 30.73 | 108.0                |  10 $f_1$   |
> | FlowTurbo     | $H_1 P_4 R_2$    | 3327      | 30.19 | 84.5      |  6 $f_1$ + 2 $f_2$    |
> | Heun's        | $H_8$            | 6667      | 28.61 | 156.2                |  16 $f_1$    |
> | FlowTurbo     | $H_1 P_6 R_3$    | 4030      | 28.60 | 104.8      |  8 $f_1$ + 3 $f_2$    |
> | Heun's        | $H_{10}$         | 8023      | 28.06 | 188.4                |   20 $f_1$   |
> | FlowTurbo     | $H_3 P_6 R_3$    | 5386      | 27.60 | 137.0       |   12 $f_1$ + 3 $f_2$   |
>
>
> These results clearly show that our method still surpasses the compiled baseline by large margins, demonstrating the effect of our velocity refiner and the pseudo corrector. We will modify these in the revised paper.
>
> **Q4: About the description of the linear interpolation.**
>
> **[Reply]** Thanks for pointing out this. We agree that linear interpolation connects noise distribution to the distribution concentrated around a particular data point in the conditional flow-matching defined in [14]. We will modify the description in L37, L89, L141 in the revision.
>
> **Q5: About the typos.**
>
> **[Reply]** Thanks for your careful reading. We will modify these typos accordingly in the revised paper.

---

> > ### Comment · Reviewer_PyDm · 2024-08-11
> > **Response to the rebuttal**
> >
> > Dear authors,
> >
> > Thank you for your time and valuable additional clarifications in the rebuttal.
> > I will keep my original positive rating.
> >
> > Best regards,
> > Reviewer

---

> > > ### Author Response · Authors · 2024-08-12
> > > **Thanks for your time and postive feedback**
> > >
> > > We greatly appreciate your response and valuable suggestions, which would improve the quality and comprehensiveness of our paper. We will update our paper accordingly in the revision.

---

### Official Review · Reviewer_yH4R · 2024-07-14

**Soundness:** 2
**Presentation:** 2
**Contribution:** 2
**Rating:** 5
**Confidence:** 4

**Summary:**

The paper presents a new approach to accelerate the sampling process in flow-based generative models. Unlike diffusion models, flow-based models, which are based on learning velocity fields through flow-matching, have not seen extensive development in efficient sampling techniques. The authors introduce FlowTurbo, a framework that utilizes a lightweight velocity refiner to estimate velocity during sampling, significantly reducing computation time while maintaining or improving image quality. Additionally, the paper proposes pseudo corrector and sample-aware compilation techniques to further enhance sampling efficiency. Experimental results demonstrate that FlowTurbo achieves substantial speedups and high-quality results in both class-conditional and text-to-image generation tasks.

**Strengths:**

1. **Technical contribution**: The introduction of a lightweight velocity refiner to approximate the offset between the velocities between adjacent timesteps seems sound and is efficient as it requires only a light-weight model.

2. **Additional Techniques**: The integration of pseudo corrector and sample-aware compilation techniques shows a well-rounded approach to improving both speed and quality of the generative process.

3. **Extensive Experiments**: The paper provides thorough experimental results, demonstrating the effectiveness of FlowTurbo across different tasks and models, including detailed comparisons with existing methods.

4. **Real-time Performance**: Achieving real-time image generation with substantial improvements in inference time without compromising quality is a notable accomplishment.

5. **Versatility**: FlowTurbo’s compatibility with various applications such as image editing and inpainting highlights its flexibility and potential for broader applications.

**Weaknesses:**

1. **Limited Comparison**: There are many efficient diffusion distillation methods, e.g., consistency distillation [1,2], consistent trajectory model [3], distribution matching distillation [4], adversarial approaches [5,6] (which also distilled using model trained with rectified flow). However, the paper has not been compared with any of those, which makes the comparison incomplete. In order to describe the technical advantage of using lightweight refiner, the author should compare with full fine-tuning (i.e., distillation) or with parameter efficient fine-tuning, e.g., LoRA. Thus, Table 2 should be filled with various baselines in order to precisely state ‘comparison with the state-of-the-arts’.
2. **Complexity of implementation**: In addition, while the method seems sound, the method is somewhat complicated to use all Heun’s method, corrector step, etc. This might be an obstacle to scale this method to a large-scale.

**References**\
[1] Song, Yang, et al. "Consistency models." arXiv preprint arXiv:2303.01469 (2023). \
[2] Song, Yang, and Prafulla Dhariwal. "Improved techniques for training consistency models." arXiv preprint arXiv:2310.14189 (2023).\
[3] Kim, Dongjun, et al. "Consistency trajectory models: Learning probability flow ode trajectory of diffusion." arXiv preprint arXiv:2310.02279 (2023).\
[4] Yin, Tianwei, et al. "One-step diffusion with distribution matching distillation." Proceedings of the IEEE/CVF Conference on Computer Vision and Pattern Recognition. 2024.\
[5] Sauer, Axel, et al. "Adversarial diffusion distillation." arXiv preprint arXiv:2311.17042 (2023).\
[6] Sauer, Axel, et al. "Fast high-resolution image synthesis with latent adversarial diffusion distillation." arXiv preprint arXiv:2403.12015 (2024).

**Questions:**

My major concern is that the paper considers improving the sampling efficiency of flow-based models which is good, but lacks comparison with other fast-sampling diffusion models. Given the theoretical or practical similarities between diffusion models and flow matching models, I believe the previous methods can be also applied to flow matching models. In that sense, there is a large gap in terms of performance, as previous works can sample within only 2-4 steps, e.g., consistency models. In order to elaborate the author’s claim that the velocity refiner is efficient and as good as those methods, the author should provide some additional comparison with those methods.

**Limitations:**

The authors mentioned some limitations that those method could not be used for diffusion models, yet it could be in a better version if they demonstrate with detailed analysis.

---

> ### Author Rebuttal · Authors · 2024-08-06
>
> We thank the reviewer for the valuable comments. We address the questions and clarify the issues accordingly as described below.
>
> **Q1: About comparisons to distillation-based methods.**
>
> **[Reply]** Thanks for your suggestions. We have detailedly discussed the difference between our FlowTurbo and distillation-based methods in our original paper (L62-L68, L100-L104, L205-L214). Although the previous diffusion distillation methods can be directly adapted to flow-based models, they do not investigate the unique properties of the flow sampling trajectory. Our method, however, aims to offer a training-efficient solution for accelerating flow-based image generation. As Reviewer PyDm suggests, the low additional training costs can make our FlowTurbo have a high impact on the acceleration of generative models.
>
> We also agree that it would be better to include more distillation methods for comparison in Table 2. We summarize the comparisons between our FlowTurbo and [1-3] on ImageNet in the following table. Note that all these methods only perform experiments on ImageNet 64x64, while our experiment setting follows SiT (ImageNet 256x256) and we cannot find a good flow-based model trained on ImageNet 64x64. The different resolution and base model choices might slightly affect the results.
>
> |Method|Resolution|FID 50K|Latency (ms / img)|Training Costs|GPU|
> |-----|-------|-----|-----|------|----|
> |CD [1]|64x64|4.70| 70.1 | 600K iters x 2048 batch size|64 A100 GPU|
> |iCT-deep [2]|64x64|2.77|70.1 | 800K iters x 4096 batch size| A100 Cluster|
> |CTM [3]| 64x64 | 1.73|70.1 | 30K iters x 2048 batch size|8 A100 GPU|
> |FlowTurbo (ours)| 256x256 |2.22 | 72.5 | 30K iters x 18 batch size| 1 A800 GPU|
>
> Besides, we also provide comparisons with more distillation methods [4-5,7-8] on text-to-image. We adopt the FID 30K on COCO as the evaluation metric. We do not include [6] since we cannot find the same evaluation metric in their paper and the models have not been released.
>
> |Method|FID 30K |Latency (s)|Training Costs|GPU|
> |-----|-----|-----|------|----|
> |LCM [7]|11.1|0.19| 100K iters x 72 batch size |8 A100 GPU|
> |LCM-LoRA [8]|23.62|0.19| - |-|
> |DMD [4]|14.93|0.09|20K iters x 2304 batch size|72 A100 GPU|
> |ADD-M [5]| 20.33 | 0.09 | 4K iters x 128 batch size| - |
> |FlowTurbo (ours)| 11.95 | 0.14| 10K iters x 16 batch size|1 A800 GPU|
>
> From the above results, we demonstrate that our FlowTurbo can achieve competitive sampling quality with previous diffusion distillation methods while requiring much fewer training costs. We will add these results to our revised paper to make the comparison clearer.
>
> [7] Luo, Simian, et al. "Latent consistency models: Synthesizing high-resolution images with few-step inference." arXiv preprint arXiv:2310.04378 (2023).
>
> [8] Luo, Simian, et al. "LCM-LoRA: A universal stable-diffusion acceleration module." arXiv preprint arXiv:2311.05556 (2023).
>
>
> **Q2: About the implementation.**
>
> **[Reply]** Thanks for the valuable comments. The motivation to adopt the velocity refiner is quite clear according to Figure 1, where we show the velocity tends to become more and more stable during the sampling. Heun's method is a well-known ODE sampler that is adopted in our baseline SiT, and the pseudo-corrector is proposed to reduce the inference costs while attaining the same convergence order as Heun's method. Therefore, the sampling paradigm of FlowTurbo is quite intuitive and reasonable. As for the scalability of our method, we have conducted experiments with SiT (675M), InstaFlow (860M), and Lumina-Next-T2I (2B) in our paper. Besides, we also conduct experiments on the provide the newly released Stable-Diffusion-3-Medium (8B) and compare our FlowTurbo with the default sampler Euler's method:
>
> |Method|FID 30K $\downarrow$|Latency (s / img)|
> |-----|-----|-----|
> |Euler| 28.4 | 2.25 |
> |FlowTurbo| 28.3 | 1.35 |
>
> The sampling quality is measured by FID 30K $\downarrow$ on MSCOCO. Besides, we also provide some qualitative comparisons on SD3 and the newly released SOTA model FLUX.1-dev (12B) in Figure A2 and Figure A3 in the attached one-page PDF, where we demonstrate that FlowTurbo can generate more visual details.
>
> We believe the results on SD3 and FLUX.1-dev, together with the experiments on various flow-based models in our original paper can demonstrate the effectiveness of our FlowTurbo at scale.
>
> **Q3: Applying FlowTurbo for diffusion models.**
>
> **[Reply]** Thanks for pointing out this. In our early experiments, we did try to transfer FlowTurbo to diffusion models. Specifically, we adopted PixArt as the base model (diffusion transformer) and tried to learn a refiner for the $\epsilon_\theta$. However, just as Figure 1 suggests, the $\epsilon_\theta$ of diffusion models changes a lot during the sampling and it is really hard to regress the offset of $\epsilon_\theta$. Alongside attempting to directly regress the offset of $\epsilon_\theta$, we have also tried to convert the $\epsilon_\theta$ to an equivalent velocity through Equation (32):
>
> $$\boldsymbol{v}=\frac{\dot{\alpha}_t}{\alpha_t}\boldsymbol{x} + (\dot{\sigma}_t-\frac{\dot{\alpha}_t\sigma_t}{\alpha_t})\boldsymbol{\epsilon},$$
> but the refiner still cannot converge. To better understand the difference between diffusion and flow-based models, we plot the curvature of the sampling trajectories in Figure A1 in the attached PDF, where we show the flow-based models enjoy lower curvature (indicating the sampling path is straighter). We will add more theoretical and empirical analysis in the revision.

---

> > ### Comment · Reviewer_yH4R · 2024-08-11
> >
> > Thank you for the rebuttal.
> >
> > It is interesting that the proposed method does not applies to diffusion model, even when changed to velocity prediction. So if we want to apply FlowTurbo for pretrained diffusion model, we have to change to flow-based model as InstaFlow did, and then apply FlowTurbo. Elaborating on this point would make the paper more valuable.
> >
> > I appreciate the authors' efforts on new experimental results for recent SOTA flow-based models. Most of my concerns were resolved and I update my rating to 5.

---

> > > ### Author Response · Authors · 2024-08-11
> > > **Thanks for raising your score and providing valuable feedback**
> > >
> > > We are glad to know that most of your concerns have been resolved. FlowTurbo is motivated by the visualization of the sampling trajectory (Figure 1), and leverages the unique property of flow-based models which have $v_t$ as a ''stable value''. As for applying FlowTurbo for pretrained diffusion models,  we think the core idea is to perform some invertible transformation $f$ of the intermediate results $x_t$ such that $f(x_t)$ is a ''stable value'', and learn a lightweight refiner to regress the offset of $f(x_t)$. We will leave this for future work to make our FlowTurbo a more general framework for the acceleration of both diffusion and flow-based models.

---

### Author Rebuttal · Authors · 2024-08-06

We sincerely thank the reviewers for the positive feedback and valuable comments on our work. In the attached one-page PDF, we provide more visualizations and qualitative results to better illustrate the motivation and effectiveness of our FlowTurbo. In Figure A1, we compare the curvature (as suggested by Reviewer RYd7) of the sampling trajectories of different models including DiT, SiT, SD3-Medium, FLUX.1-dev, and Open-Sora. In Figure A2-A4, we provide more qualitative comparisons on SD3-Medium (a SOTA flow model by StabilityAI, released on June 12th), FLUX.1-dev (a SOTA flow model by Black Forest Labs, released on August 3rd), Open-Sora (a video-generation flow model). These qualitative clearly demonstrate our FlowTurbo generates visual content with higher fidelity and more details than the default sampler.

---

### Decision · Program_Chairs · 2024-09-25

**Decision:**

Accept (poster)

**Comment:**

The paper proposes several improvements to sampling of flow-based models, including lightweight velocity refiner and "pseudo corrector". These together improve the quality-speed tradeoff of sampling, resulting in SOTA quality/speed frontier.

The paper is clearly written, the proposed modifications make sense, the experiments are thorough, and the results are good. All reviewers recommend acceptance. I concur and recommend acceptance too.